



**InundatEd: A Large-scale Flood Risk Modeling System on a Big-data -**
**Discrete Global Grid System Framework**
Chiranjib Chaudhuri[1], Annie Gray[1], and Colin Robertson[1]
[1]Wilfrid Laurier University, Department of Geography and Environmental Studies,
Waterloo, Canada
Email: chiranjibchaudhuri@gmail.com
Keywords: Flood modeling system, Height Above Nearest Drainage, Discrete
Global Grid System, IDEAS, Web-GIS, R/Shiny, Manning's Equation, Regional
Regression.





**Abstract**
Despite the high historical losses attributed to flood events, Canadian flood mitigation efforts have
been hindered by a dearth of current, accessible flood extent/risk models and maps. Such resources
often entail large datasets and high computational requirements. This study presents a novel,
computationally efficient flood inundation modelling framework ("InundatEd") using the height
above nearest drainage-based solution for Manning's equation, implemented in a big-data discrete
global grid systems-based architecture with a web-GIS platform. Specifically, this study aimed to
develop, present, and validate InundatEd through binary classification comparisons to known flood
extents. The framework is divided into multiple swappable modules including: GIS pre-
processing; regional regression; inundation model; and web-GIS visualization. Extent testing and
processing speed results indicate the value of a DGGS-based architecture alongside a simple
conceptual inundation model and a dynamic user interface.





## Introduction:

Globally from 1994 to 2013 flood events accounted for 43% of recorded natural disasters (Centre for Research on the Epidemiology of Disasters, 2016). Flooding is responsible for one third of natural disaster costs in Europe (Albano, Sole, Adamowski, Perrone, & Inam, 2018), while in Canada mean annual losses of $1-2 billion (CAD) are attributed to flood disasters (Oubennaceur et al., 2019). A 2013 flood in southern Alberta, costing over 1.7 billion dollars (CAD) in insured property damages, is the most expensive natural disaster in Canadian history (Stevens & Hanschka, 2014). Rapid economic development and urbanization during the last few decades – particularly urban development in close proximity to Canadian waters following population expansions of the 1950s-1960s – have increased the amount of exposure and in-turn the economic damages of flood events (Robert et al., 2003). Despite increasing risks and impacts of flood events, many continue to settle in flood-prone areas, making the availability of accurate, timely, and detailed flood information a critical information need (Pal, 2002).

Mitigating the considerable economic impact of flood events; the design of effective emergency response measures; the sustainable management of watersheds and water resources; and flood risk management, including the process of public flood risk education, have long been informed by the practice of flood (inundation) modelling, which aims to understand, quantify, and represent the characteristics and impacts of flood events across a range of spatial and temporal scales (Handmer, 1980; Stevens & Hanschka, 2014; Teng et al., 2017, 2019; Towe et al., 2020). Flood inundation modelling research has increased in response to such factors as predicted climate change impacts (Wilby & Keenan, 2012) and advancements in computer, GIS (Geographic Information Systems), and remote sensing technologies, among others (Kalyanapu, Shankar, Pardyjak, Judi, & Burian, 2011; Vojtek & Vojteková, 2016; Wang & Cheng, 2007). Flood modelling approaches can be broadly divided into three model classes: empirical; hydrodynamic; and simplified/conceptual. Empirical methods entail direct observation through methods such as remote sensing, measurements, and surveying, and have since evolved into statistical methods informed by fitting relationships to empirical data. Hydrodynamic models, incorporating three subclasses (one-dimensional, two-dimensional, and three-dimensional), consider fluid motion in terms of physical laws to derive and solve equations. The third model class, simple conceptual, has become increasingly well-known in the contexts of large study areas, data scarcity, and/or stochastic modeling and encompasses the majority of recent developments in inundation modelling





practices. Relative to the typically complex hydrodynamic model class, simple conceptual models
simplify the physical processes and are characterized by much shorter processing times
(Oubennaceur et al., 2019; Teng et al., 2017, 2019). While each class has contributed substantially
to the advancement of flood risk mapping and forecasting practices, a consistent barrier has been
the trade-off between computer processing time and model complexity (Neal, Dunne, Sampson,
Smith, & Bates, 2018), especially with respect to two-dimensional and three-dimensional
hydrodynamic models, which entail specialized expertise to derive and apply physical and fluid
motion laws, require adequate data to resolve equations, and the computational resources to
process the equations. Neal et al. (2018) summarized the proposed solutions to such challenges as
relating to 1) modifications to governing equations or 2) code parallelization, with the latter
informing the method proposed in Oubennaceur et al. (2019). With respect to 2D/3D
hydrodynamic model code parallelization, Vacondio et al. (2017) listed two approaches: classical
(Message Passing Interface) and Graphics Processing Units (GPUs). The GPU-accelerated method
has been shown to decrease execution times, whilst avoiding the use of supercomputers, for high-
resolution, regional-scale flood simulations (e.g., Ferrari et al. (2020), Vacondio et al. (2017),
Wang & Yang (2020), and Xing et al. (2019)). However, the GPU-accelerated method is still
limited in terms of the hardware requirement (graphics cards), the use of uniform and/or non-
uniform grids (Vacondio et al. (2017)), and the need for specific, specialized modelling programs
to handle the input data required to solve complex hydrodynamic equations. The ongoing
development of simple conceptual inundation models offers another avenue to handle limitations
such as computation requirements and data scarcity, allowing areas poorly served by standard
hydrodynamic modeling, to be provided with up-to-date flood extent maps and provided with
platforms with which the public can view and interact with the simulated floods (Tavares da Costa,
2019). Although simple conceptual models using such methods as linear binary classification and
Geomorphic Flood Index (Samela et al., 2017, 2018) have been, and continue to be, developed,
the combination of simple conceptual flood methods with big-data approaches remains largely
uninvestigated (Tavares da Costa, 2019).

96        Recent advances in big data architectures may hold potential to retain enough model

complexity to be useful while providing computational speedups that support widespread and
system agnostic model development and deployment. There is an increasing need for examination
of the potential of decision-making through data-driven approach in flood risk management and

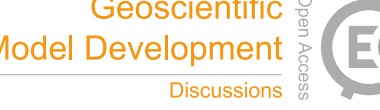

investigation a suitable software architecture and associated cohort of methodologies which
involves more data-centric architecture (Towe et al., 2020). Discrete global grid systems (DGGS)
are emerging as a data model for a digital earth framework (Craglia et al. 2012; Craglia et al.,
2008). One of the more promising aspects of DGGS data models to handle big spatial data is their
ability to integrate heterogeneous spatial data into a common spatial fabric. This structure is
suitable for rapid model developments where models can be split into unit processing regions.
Furthermore, with the help of DGGS the model can be ported to a decentralized big-data
processing system and many computations can be scaled for millions of unit regions. A recently
developed DGGS-based data model and modelling environment called an Integrated Discrete
Environmental Analytics System (IDEAS) is one such system which implements a multi-
resolution hexagon tiling data structure within a hybrid relational database environment
(Robertson, Chaudhuri, Hojati, & Roberts, 2020). Notably, and in contrast to previous systems,
the only special installation entailed by IDEAS is a relational database. The system exploits the
hardware capability of the database itself which can potentially incorporate the following: GPU(s),
distributed storage, and a cloud database. In this paper we employ the IDEAS framework for the
efficient computation, simulation, analysis, and mapping of flood events for risk mitigation in a
Canadian context.
In Canada, nationwide flood mapping efforts were catalyzed by extensive flood damages
to southern Ontario due to Hurricane Hazel in 1954, resulting in the Canadian government's
institution of the National Flood Damage Reduction Program (NFDRP) in 1975 (Burrell & Keefe,
1989). The NFDRP, a joint federal/provincial undertaking, entailed a number of co-signed
agreements related to the reduction of risks of human suffering, loss of life, of assistance costs,
and the limitation of flood mitigation infrastructure (Robert et al., 2003). The program set the stage
for the creation of high quality flood risk maps as a medium to provide information to the public,
to inform land use zoning, and to inform disaster response strategies, among other goals (Handmer,
1980), and demonstrated the need for and value of effective Canadian flood mapping practices.
Regrettably, the program was slowly phased out and terminated by 1996 (Pal, 2002). Flood
mapping responsibilities previously encompassed by the program were delegated to various levels
of government, resulting in a heterogeneous set of mapping standards and practices which still
hinder Canadian flood management practices today (Calamai & Minano, 2017). Moreover, best
practices in flood hazard mapping are rarely made freely available to the Canadian public.





Flood risk maps as decision support tools can build the capacity of individuals to make
informed and sustainable investment and residence decisions in an age of climate concern and
environmental change (Albano et al., 2018). The current state of public knowledge of flooding
risks is unsatisfactory, with an estimated 94% of 2300 Canadian respondents in highly flood-prone
areas lacking awareness of the flood-related risks to themselves and their property, per a 2016
national survey (Calamai & Minano, 2017; Thistlethwaite, Henstra, Brown, & Scott, 2018;
Thistlethwaite, Henstra, Peddle, & Scott, 2017). Calls for better transparency and access to reliable
flood risk maps and data with which to improve public awareness and understanding of flood risks
is in line with a contemporary trend toward more open and reproducible environmental models
(Gebetsroither-Geringer, Stollnberger, & Peters-Anders, 2018). There is an opportunity to utilize
big data architectures and recent developments in flood inundation modelling and risk assessment
technologies to make flood risk information more accessible.
The aim of this paper is threefold: 1) propose a simple conceptual inundation model
implemented in big-data architecture; 2) test the model and its results through comparison to
known extents of previous flood events; and 3) present the resultant flood maps via an open source,
interactive web application.

**2. Methods**

**2.1 Overview**
The modelling component of InundatEd incorporated four general stages: 1) GIS pre-processing;
2) flood frequency analysis and regional regression; 3) the application of the catchment integrated
Manning's Equation; 4) the application of FEMA's Hazus Depth-Damage functions; and 5)
upscaling the model to a discrete global grid systems data model. Sections 2.2.1 to 2.2.5 describe
stages 1-5 respectively.
The second component of InundatEd's development was the design of a Web-GIS
interface, described in Section 2.3, which liaises with and between the big data architecture, the
flood models' outputs as defined by user inputs, and FEMA's Hazus depth-damage functions
(Nastev & Todorov, 2013). Section 2.4 subsequently links the Web-GIS interface conceptually to
previous sections by providing a summary of InundatEd's system structure and its operation.
Finally, simulated flood extents using InundatEd's methodology were compared to the extents of





observed, historical flood extent polygons within the Grand River watershed and the Ottawa River
watershed, provided respectively by the Grand River Conservation Authority and Environment
Canada. The comparison and testing process is described in Section 2.5.


**2.2. Modelling**

2.2.1 – Stage 1: GIS Pre-processing
The following GIS input data were obtained from Natural Resources Canada for the Grand River
and Ottawa River watersheds and cropped to their respective study area: Digital Elevation Models
(Canada Centre for Mapping and Earth Observation, 2015); river network vector shapefiles
(Strategic Policy and Innovation Centre, 2019); and Land Use Land Cover (LULC) (Canada
Centre for Remote Sensing, 2019). Figure 1 shows the input Digital Elevation Model data, with
elevation values given in metres with reference to the CGVD2013 vertical datum. The remaining
GIS input data is shown in Supplementary Figure S1. Very small networks, independent of the
higher-order channels, were deleted from both regions. ArcGIS Desktop's Raster Calculator tool
was used to burn the river network vector into the DEM in preparation for further analysis.
TauDEM (Terrain Analysis Using Digital Elevation Models) (Tarboton, 2005), an open-source
tool for hydrological terrain analysis, was then used to determine drainage directions and drainage
accumulation  (Tarboton & Ames, 2004) within the watersheds of interest. Each watershed's
drainage network was then established in TauDEM by defining a minimum threshold of two square
kilometres on the contributory area of each pixel for the Grand River watershed and ten square
kilometres for the Ottawa River watershed. Separately, a value of Manning's n was determined for
each 30 x 30 metre pixel of the study areas based on land use/ land cover attributes (Comber &
Wulder, 2019). To this end, the input LULC classes (Canada Centre for Remote Sensing, 2019)
within the study watersheds were mapped to the nearest class of the similar land cover classes
documented in Chow (1959, Table 5-6) and Brunner (2016, Figure 3-19), from which the
respective values of Manning's N were used. Table 1 provides the utilized input LULC classes,
their respective description provided by NRCAN, and the employed n values. Height Above
Nearest Drainage (HAND) (Rahmati, Kornejady, Samadi, Nobre, & Melesse, 2018; Garousi-
Nejad, Tarboton, Aboutalebi, & Torres-Rua, 2019) was also calculated in TauDEM with reference





to the DEM and derived drainage network. Figure 2 provides a visual accounting of this stage of
the modelling component.

2.2.2. Stage 2: Regional Regression and Flood Frequency Analysis
The index flood approach - a regional regression model based on annual maximum discharge
data (Darlymple, 1960) and described in Hailegeorgis & Alfredsen (2017)- was used to derive
the discharges by return period at sub-catchment outlets. The model includes two sections: a) a
relationship between index flood and contributory upstream area for each hydrometric station
and each subcatchment outlet (regional regression); and  b) a flood frequency analysis to
estimate the quantile values of the departures,with a departure defined as discharge at given
station divided by the index flood of that same station). The index flood approach entails the
following assumptions: a) the flood quantiles at any hydrometric site can be segregated into two
components – an index flood and regional growth curve (RGC) -; b) the index flood at a given
location relates to the (sub)catchment characteristics via a power-scaling equation, either in a
simpler case which considers only upstream contributory area or in a more complex case which
incorporates land use/ land cover, soil, and climate information; and c) within a homogeneous
region the departure/ratio between the index flood and discharge at hydrometric sites yields a
single regional growth curve which can relate the discharge and return period.

Per assumption a, the index flood at each hydrometric station is required. To this end, annual
maximum discharge values ($m^3s^{-1}$) were extracted within R (R Core Team, 2019) at hydrometric
stations maintained by Environment Canada within the Grand River and Ottawa River watersheds
(HYDAT) (Hutchinson, 2016). Only stations with a period of record >= 10 years of annual
maximum discharge were maintained (n = 32 and n = 54, respectively). The minimum and
maximum periods of record for the Grand River watershed were 12 years and 86 years,
respectively. Periods of record for the Ottawa River watershed ranged from a minimum of 10 years
to a maximum of 58 years. A median annual maximum discharge value was then calculated ($\tilde{Q}$)
for each hydrometric station. As discussed in Hailegeorgis & Alfredsen (2017), although the index
flood is generally the sample mean of a set of annual maximum discharge values, index floods
have also been evaluated based on the sample median (eg. Wilson et al., 2011) at the suggestion
of Robson & Reed (1999).  Finally, the index flood values ($\tilde{Q}$) were used to normalize the observed
annual maximum discharge values (Q) at their respective station ($q_i = Q/ \tilde{Q}$).




With respect to regional regression and assumption b of the index flood method, a generalized
linear model was applied to relate $\log_{10}$ transformed $\tilde{Q}$ values to $\log_{10}$ transformed upstream area
values at each hydrometric station. The generalized linear model assumed an ordinary least squares
error distribution. The results of the generalized linear model for each watershed allowed for the
calculation of previously unknown $\tilde{Q}$ values for each subcatchment outlet. In a more complex
model (Fouad et. al. 2016), other catchment characteristics such as land use/land cover, geology,
etc. could be used. However, in the case of the proposed model the correlations between the
calculated and observed index floods, on the sole basis of discharge records and a linear model
relating upstream area, were high as discussed in the Results section. Thus, the simpler method
was used to estimate index floods and to relate index flood to contributory area at hydrometric
stations and subcatchment outlets. Thus, the regional regression model derived a relationship
between index flood ($\tilde{Q}$) and upstream contributory area for each hydrometric station i or
subcatchment outlet. The relationship between index flood at station i or at a subcatchment outlet
($\widetilde{Q^i}$) and upstream contributory area ($A_i$) is given by:
$$\tilde{Q}^i = aA_i^c \quad (1)$$
where $a$ is the index flood discharge response at a unit catchment outlet (or at a hydrometric
station) and $c$ is the scaling constant. We took the logarithm of Equation (1) on both sides - a
procedure used in noted in Hailegeorgis & Alfredsen (2017) as used in Eaton, Church, & Ham
(2002) - yielding a linear relationship which was solved using the Ordinary Least Squares approach
(Haddad et al. (2011).
The selection of a suitable probability distribution model – a common tool in hydrologic modelling
studies (Langat et al., 2019; Singh, 2015)- for use in a watershed where the flow has been modelled
due to human abstraction is a fundamental step of the analysis process and must account for
disturbance-related changes to the extreme value characteristics of the flow. While solutions to
this problem have been proposed in the literature, artificial abstraction fundamentally changes the
extreme value characteristics of the flow, thereby hindering the usability of most distributional
forms (Kamal et. al. 2017). Many researchers have tried to address this problem by putting explicit
assumptions on types of non-stationarity affecting the river discharge and are able to devise a
closed mathematical formulation which enables the parametric distributions to handle such non-
stationarity. However, such methods typically entail knowledge of the specific design return
periods of individual flood prevention structures (Salas & Obeysekera, 2014), many of which are





absent in our case. To circumvent this problem, we used a non-parametric approach for the regional
growth curve (RGC), which requires no fundamental sample characteristics. Thus, modified flood
records and limited information notwithstanding, flood frequency estimation is possible using the
index flood approach. Per assumption c of the index flood method, a log-spline non-parametric
approach was taken to model a RGC (Stone, Hansen, Kooperberg, & Truong, 1997) for each study
watershed. Specifically, the index flood values ($\tilde{Q}$) were used to normalize the observed annual
maximum discharge values (Q) at their respective station ($Q_i = Q/\tilde{Q}$). The $Q_i$ values (n= 1487 and
n = 1248 for the Ottawa River watershed and the Grand River watershed, respectively) were then
fitted to a logspline distribution for their respective watershed. The discharge quantiles ($Q_r$) were
extracted for the following return periods (T, years):  1.25, 1.5, 2.0, 2.33, 5, 10, 25, 50, 100, 200,
and 500. The return periods were first converted to a cumulative distribution function:
$$\text{CDF} = 1 - \left(\frac{1}{T}\right) \quad (2)$$

Finally, flood quantile estimations were calculated for each return period as shown below:
$$Q_T^i = \widetilde{Q^i} q_T \quad (3)$$

such that T is a specified return period in years;  $Q_T^i$ is a quantile estimate of discharge for the
specified return period T (years) at a specified station i (or a subcatchment outlet); $\widetilde{Q^i}$ is the "index
flood" at the same station i (or at the same subcatchment outlet); i = 1,2,…,N where N =32 for the
Grand River watershed or N= 54 for the Ottawa River watershed; and $q_T$ is the regional growth
curve as described above. Figure 3 provides a visual accounting of the regional regression and
flood frequency analysis methodology described in this section.
2.2.3 Stage 3: Catchment Integrated Manning's Equation
Manning's formula (Song et. al., 2017) is widely used to calculate the velocity and subsequently
the discharge of any cross-section of an open channel. The Manning's equation is given in SI units
by:
$$Q = \frac{1}{n} R_h^{\frac{2}{3}} A S^{\frac{1}{2}} \quad (4)$$

such that Q is discharge in cubic metres per second, A represents the cross-sectional area, n is a
roughness coefficient, $R_h$ is the hydraulic radius, and S represents slope (fall over run) along the
flow path. Despite its widespread use, robustness, and relative ease of use, Manning's Equation
has an inherent problem which comes from the uncertain orientation of cross-sections. To mitigate
this problem, we integrated Manning's Equation along the drainage lines within the catchment,





accounting for the slope of each grid cell to yield bed area and derived the stage-discharge
relationship. This strategy uses hydrological terrain analysis, discussed previously in Section 2.2.1,
to determine the Height Above Nearest Drainage (HAND) of each pixel (Rodda, 2005; Rennó et
al., 2008). The HAND method determines the height of every grid cell to the closest stream cell it
drains to. In other words, each grid cell's HAND estimation is the water height at which that cell
is immersed. The inundation extent of a given water level, can be controlled by choosing all the
cells with a HAND less than or equal to the given level. The water depth at every cell can then be
calculated as the water level minus the HAND value of the corresponding cell. The relevance of
HAND to the field of flood modelling has been demonstrated in the literature (Rodda, 2005, Nobre
et al., 2016). Its documented use notwithstanding, HAND's potential applications to the depiction
of stream geometry information and to the investigation of stage-discharge connections have not
been well investigated. Hydraulic methods of discharge calculation typically entail hydraulic
parameters derived from the known geometry of a channel. In contrast, the HAND method does
not require channel geometry to determine hydraulic parameters.

The conceptual framework for implementing HAND to estimate the channel hydraulic properties
and rating curve is as follows: for any reach at water level h, all the cells with a HAND value < h
compose the inundated zone F(h), which is a subarea of the reach catchment. The water depth at
any cell in the inundated zone F(h) is the difference between the reach-average water level h and
the HAND of that cell, $HAND_c$, which can be represented as: depth = $HAND_c$-h. Since a uniform
reach-average water level h is applied to check the inundation of any cell within the catchment,
the inundated zone F(h) refers to that reach level. The water surface area of any inundated cell is
equal to the area of the cell $A_c$. This case study uses 30 metre x 30 metre grid cells, thus in this
case $A_c = 900$ m$^2$. The channel bed area for each inundated cell is given by
$$A_s = A_c \sqrt{(1 + slope^2)} \quad (5)$$
where slope is the surface slope of the inundated pixel expressed as rise over run or inverse tangent
of the slope angle. This equation approximates the surface area of the grid cell as the area of the
planar surface with surface slope, which intersects with the horizontal projected area of the grid
cell. The flood volume of each inundated pixel at a water depth of h can be calculated as $V_c$ (h)=$A_c$
(h-$HAND_c$). If the reach length L is known, the reach-averaged cross section area for each pixel is
given by $A_i$=$V_c$/L. Similarly, the reach-averaged cross section wetted perimeter for each inundated



pixel $P_i(h) = A_s/L$. Therefore, the hydraulic radius for each inundated pixel is given by $R_i = A_i/P_i$.
Therefore, we can estimate the reach-averaged cross-section area $A = \sum_i A_i$, perimeter $P = \sum_i P_i$,
and hydraulic radius $R = A/P$ for the entire flooded area. The composite Manning's n is estimated
using the Lotter method (Tullis, 2012) and is given by:
$$n = \frac{PR^{\frac{5}{3}}}{\sum_i \frac{1}{n_i} P_i R_i^{\frac{5}{3}}} \quad (6)$$

Thus the discharge Q(h) corresponding to inundation height can be computed by the Manning's
equation and given by:
$$Q(h) = \frac{1}{n} R^{\frac{2}{3}} A S^{\frac{1}{2}} \quad (7)$$

where S is the slope of the river. Figure 4 displays the sequence of methods outlined for the
Catchment Integrated Manning's Equation method.

2.2.4 Stage 4: Damage Computation
To contextualize the modelled inundation depths, FEMA's Hazus Depth-Damage functions were
applied to the calculated depths via the R package Hazus (https://www.fema.gov/hazus) (Goteti,
2014). Using the Hazus package, estimated percentage losses can be generated for model output
inundation depths at individual locations specified by the user. Furthermore, the Hazus loss
percentages are contingent on building-specific properties, offering a built-in variety of building
types, descriptions, and situations (e.g., fresh water vs. salt water) to tailor final estimations to a
user's personal experience. The use of Hazus within the R Development environment allows for
seamless integration with a user interface for inputs such as building type.

2.2.5 Stage 5: Upscaling and Data Conversion
The proposed InundatEd inundation model simulates the flood-depth distributions for each
catchment independently. This makes this model suitable to be ported to a DGGS-based data
model and processing system. Following the GIS preprocessing, done in TauDEM as discussed in
Section 2.2.1, the required data was converted to a DGGS representation, as outlined in Robertson
et al., (2020). Supplementary Figure S2 for raster input data (S2a), polygon (vector) input data
(S2b), and network (directional polyline vector) input data (S2c). For raster data (S2a), the
bounding box is used to extract a set of DGGS cells, and then for each DGGS cell's centroid the


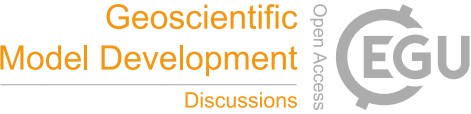

raster value is extracted. To convert polygon data to a DGGS data model, we sample from its
interior and its boundary separately using uniform sampling. Then each sample point is converted
into DGGS cells based on its coordinates and stored into IDEAS data model by aggregating both
sets of DGGS cells (Figure S2b). The same process for the border extraction is applied to the
polylines and networks, however with network data the order of the cells is also stored as a flag to
use in directional analysis (Figure S2c). Following conversion, the data was ported to a 40-node
IBM Netezza Database for subsequent calculations. General, systematic limitations of the
InundatEd IDEAS-based inundation model are discussed in Section 3.1.

**2.3 Web-GIS Interface**
The R/Shiny platform and the R-Studio development environment were used to design the user
interface and server components of an online web application, allowing users to query and interact
with the inundation model. Features of R specific to InundatEd's modelling workflow were its
support of the Hazus damage functions and its support for DGGS spatial data. Shown in Figure
5a, the InundatEd user interface offers widgets for the following user inputs: address (text);
discharge (slider); and return period (dropown), as well as tabs for viewing interactive graphs. The
InundatEd user interface also features an interative map which leverages the Leafgl R package
(Appelhans & Fay, 2019) for seamless integration with the DGGS data model. Users may click on
the map to obtain point-specific depth information, which can be passed to the Hazus damages
computation.

**2.4 InundatEd Flood Information System – System Structure Summary**
Figure 5b displays the overall system structure and linkages for the InundatEd flood information
system. GIS input data, as discussed in Section 2.2, were staged, pre-processed, and ported to the
database. Data querying was used to compute 'in-database' inundation (flood depth) and related
damages (methods outlined in Section 2.1) in response to user interface inputs to the R/Shiny UI.

**2.5 Flood Data Comparison and Model Testing**
2.5.0 Study Areas





As preliminary testing domains, we created flood inundation models for the Grand River Basin
and Ottawa River Basin respectively, both located in Ontario, Canada. Each basin has experienced
historical flooding and have implemented varying measures of flood control. Table 2 shows
different salient characteristics of these catchments. For the purposes of graphing and discussion
of station-specific period of record (number of years with a recorded annual maximum discharge)
on theoretical vs estimated flood quantiles, two stations from each study watershed were selected,
one each for high period of record and low period of record. For the Grand River watershed,
stations 02GA003 and 02GA047 were selected for high and low period of record, respectively.
For the Ottawa River watershed, stations 02KF006 and 02JE028 were selected, respectively.
"Theoretical quantiles" are here defined as the quantiles generated by our model based on the
logspline fit, which incorporates annual maximum discharge values from multiple stations across
each study watershed (Section 2.2.2 and Figure 3). In contrast, "estimated quantiles" are here
defined as the flood quantiles calculated simply by extracting the quantiles for the desired return
periods from the raw annual maximum discharge values observed at the hydrometric station of
interest.
2.5.1. Ottawa River Watershed
Four flood extent polygons (FEPs) provided by Natural Resources Canada (Natural Resources
Canada, 2018, 2020) from the May-June 2019 flood season were used as "observed" floods to test
the model outputs for the Ottawa River watershed. Each FEP represented a previously digitized
floodwater extent at a specified date/time.
A second criterion for selection was that the hydrometric station(s) intersected by the FEP provided
discharge data for the FEP's respective datetime. Two hydrometric stations which met both criteria
were selected: 02KF005 and 02KB001. The following procedure was followed for each FEP using
the corresponding hydrometric station (02KF005 or 02KB001), the station level index flood ($\tilde{Q}$,
previously calculated during Section 2.2.2), and the observed discharge ($Q_{obs}$). In both cases, the
logspline fit for the Ottawa River watershed, previously generated during Section 2.2.2, was also
used.



The observed discharge ($Q_{obs}$) was divded by the corresponding hydrometric station's index flood
($\tilde{Q}$) ($Q_i = Q_{obs} / \tilde{Q}$) The cumulative probability of $Q_i$ was converted to a return period using the
following equation:
$$\text{return period (years)} = \frac{1}{1 - cumulative\ probability} \quad (8)$$
To generate each simluated flood for comparison to its observed counterpart, the methodology
outlined in Sections 2.2.2 and 2.2.3 was repeated with the four new return periods appended to
the original list of return periods in Section 2.2.2. Table 3 lists each FEP, the corresponding
intersected hydrometric station, the period of record used for each station to calculate $\tilde{Q}$, the
observed discharge, the resultant cumulative probability value, and the final return period used to
generate each simulated flood.

2.5.2. Grand River Watershed
Regulatory floodplain extent data (the greater of RP=100 or discharge from Hurricane Hazel,
"observed" flood extent) was obtained from the Grand River Conservation Authority (GRCA)
(Grand River Conservation Authority, 2019). However, analysis revealed that, at most hydrometric
stations in the Grand River wateshed, the 100-year return period yielded higher discharge values
relative to the "Hurricane Hazel" storm. Thus, the 100-year return period could be used.  The
estimated flood extent for RP=100 was generated per sections 2.2.1-2.2.3. Table S1 provides a
discharge comparison between the 100-year return period and the regulatory storm.

2.5.3. Flood Extent Comparisons
For both the Grand River watershed and the Ottawa River watershed, only those subcatchments
in close proximity to the observed flood extent polygons were retained for visualization
purposes. To this end, a criterion was applied to subcatchments in the Grand River watershed
requiring an intersection with the observed flood polygon of >= 20% of the subcatchment's area.
For the Ottawa River watershed, due to the use of station-specific observed discharges, an
additional criterion was applied: that a given subcatchment intersects with a network line with
contributory upstream area >= 80% and contributory upstream area <= 120% of the observed
upstream area of the hydrometric station (02KF005 or 02KB001). Table S2 provides by-
subcatchment areas of the observed flood extent polygons whose subcatchments were eliminated
based on the 20% intersection threshold. Per Table S2, one excluded subcatchment (10505) had




an intersection value >= 20%, attributable in part to the presence of a tributary along which it
was not expected that the return period would be properly scaled but which intersected the
subcatchment. Additionally, due to the pluvial nature of the flooding in that subcatchment, it was
once again expected that the return period as a function of the river discharge would not be
properly scaled without the presence of a hydrometric station to provide discharge information.

Binary classification metrics have been used to compare between observed and simulated floods
in cases where the focus is on extent, not depth (eg Papaioannou et al., 2016; Wing et al., 2017;
Chicco & Jurman, 2020). A binary classification (or 2x2 contingency) method was used to
compare the simulated flood extent rasters to the extents of their observed counterparts, whereby
a confusion matrix was generated for each subcatchment. Multiple accuracy measures were
calculated from the contingency tables to support the evaluation of the flood model, including:
True Positive Rate (TPR). True Negative Rate (TNR), Accuracy, Matthews Correlation
Coefficient (MCC) (Chicco & Jurman, 2020), and the Critical Success Index (CSI) ( e.g.,
Papaioannou et al, 2016; Stephens & Bates, 2015). The MCC is a summary measure of a
confusion matrix which is robust to differences in abundance in classes. Matthews Correlation
Coefficient (MCC) is defined as:
$$MCC = \frac{TP \; x \; TN - FP \; x \; FN}{\sqrt{(TP+FP)(TP+FN)(TN+FP)(TN+FN)}} \quad (9)$$
Such that TP = true positive, TN = true negative, FP = false positive, and FN = false negative.
**3. Results and Discussion**
**3.1 Model Processes and DGGS**
Intermediate model outputs for the Grand River and Ottawa River watersheds - Height Above
Nearest Drainage, delineated river networks, and Manning's n- are displayed in Figure 6.
Figure 7 visualizes results for the Grand River watershed and for the Ottawa River watershed for
the following method components: calculation of hydrometric station upstream (contributory)
area; index flood regression as represented by the correlation of logged index discharge and logged
upstream area; and flood frequency as represented by discharge against a Gumbel transformed
return period (years), for the stations respectively representative of high and low observations.
Figures 7a and 7b plot the log of calculated upstream area against the log of observed upstream



area, yielding respective Pearson correlation coefficients of 0.99 and 0.63 for the Grand River and
Ottawa River watersheds. The difference in correlation quality can be accounted for in part by the
difference in the relative complexities of the delineated networks of the Grand River and Ottawa
River watersheds. With respect to regional regression, Figure 7c visualizes the relationship
between predicted index flood discharge and contributory upstream area, at individual hydrometric
stations, for the Grand River and Ottawa River watersheds (R = 0.83 and 0.95, respectively).The
regional growth curves for both the Grand River watershed and the Ottawa River watershed are
shown in Figure 7d. To compare the proposed approach of using log-spline distribution against a
traditional parametric distribution we fitted a Generalized Extreme Value (GEV) distribution to
the RGC (Supplementary Figure S3). With respect to the log-spline RGCs, AIC values of 1861.69
and 867.69 and (-2)(logliklihood) values of 1826.04 and 809.26 were reported for the Grand River
watershed and Ottawa River watershed respectively. The log-spline (-2)(logliklihood) values were
lower than their GEV counterparts (1837.56 and 880.12) for both watersheds. For the Ottawa River
watershed, the log-spline AIC value, 867.69, was also lower than that of its GEV counterpart
(886.12).  Furthermore, the use of the log-spline distribution allows for a consistent method which
can be applied readily across any watershed without careful calibration of the distribution function.
Thus, the log-spline distribution was used for the regional growth curves. The lower values of the
normalized discharge shown in Figure 7d for higher return periods (2-3) for the Ottawa River
watershed suggest relatively more structural alternations within the watershed, for instance flood
control and dams, than the Grand River watershed (Ottawa Riverkeeper, 2020). The Grand River
watershed yielded relatively higher values of normalized discharge (>3) at higher return periods
in Figure 7d. Figure 8 shows the comparison of estimated flood quantiles against theoretical flood
quantiles at individual stations from both study watersheds for cases of high and low observation
counts, such that "discharge count" refers to the number of years for which an annual maximum
discharge was recorded (period of record). Return periods (T, years) have been converted in terms
of the Gumbel reduced variable as follows:
$$Gumbel = -ln\left[ln\left(\frac{T}{T-1}\right)\right] (10)$$
As expected, for the stations with high observation counts (n = 101 and n = 84 for the Grand River
watershed (Figure 8a) and Ottawa River watershed (Figure 8b), respectively) the theoretical and
estimated return periods are closer, at least for lower return periods. The value of long periods of
record can also be considered in terms of the 5T threshold (shown as the dotted lines in Figure 8).





The 5T threshold requires that, for the reasonable estimation of a quantile for a desired return
period T, there be at least 5T years of data (Hailegeorgis & Alfredsen, 2017).

The major limitations of this model stem from the nascent stage of the IDEAS geo-data model and
the exclusion of hydrological processing algorithms. The initial offline GIS-processing entailed
lengthy input data conversions to the IDEAS system prior to subsequent calculations. Furthermore,
in contrast to the square raster where we have two orthogonal axis, the hexagonal cells in the
IDEAS data model consists of a reference system of 3 non-orthogonal axis which makes the
computation of the essential hydraulic parameters such as drainage direction and slope quite
different from the traditional square raster system. Thus, GIS pre-processing computed on a square
raster doesn't essentially hold true in case of IDEAS's hexagonal gridding system wherein
subsequent calculations were performed, meriting additional development and testing.

### 3.2 Web-GIS Interface

A pre-alpha version of the InundatEd app is available at https://spatial.wlu.ca/inundated/. Source
code for the most recent version of InundatEd will be publicly available on GitHub (Spatial Lab,
2020). The use of R/Shiny to develop InundatEd and its provision on GitHub encourages
transparency, ongoing development, and response to user feedback and preferences.

### 3.3 Model Testing


The following return periods (in years) were observed for FEPs intersecting hydrometric station
02KF005 in the Ottawa River watershed: 26.5, 16.52, and 25.96. Additionally, a return period of
42.69 years was observed for a FEP intersecting hydrometric station 02KB001 in the Ottawa River
watershed. The 100-year return period was tested for the Grand River watershed. Binary
classification results for the Grand River watershed are shown in Figure 9 for four comparison
metrics: Matthews Correlation Coefficient, Accuracy, True Positive Rate, and True Negative Rate.
Figure 10 presents Matthews Correlation Coefficient and Accuracy results for the four Ottawa
River watershed cases, with True Positive and True Negative results presented in Supplemetary
Figure S4. Although the results for both the Grand River watershed and the Ottawa River
watershed suggest substantial agreement between the respective observed and simulated flood
extents, a number of considerations, including input data characteristics and metric bias, require





that the presented results be taken with caution and, in some cases, offer clear paths for
improvement. With respect to input data, the simulated floods presented within this case study are
limited by the initial use of a 30m x 30 DEM raster. As concluded by Papaioannou et al. (2016),
floodplain modelling is sensitive to both the resolution of the input DEM and to the choice of
modelling approach.

As noted in Lim & Brandt (2019), the reliability of the observed flood extent polygons also merits
comment. In this case study, the observed FEPs for the Ottawa River watershed were originally
digitized from remotely sensed data and thus carry forward the errors and uncertainties from prior
processing. The Grand River watershed's 100-year return period extent was also generated outside
of this study and potentially carries multiple sources of error and uncertainty. However, evaluation
of the exact extent to which errors present in the observed flood extent polygons could have
impacted the binary classification results was not an objective of this study.

With respect to the binary classification metrics for both watersheds, the generally high Accuracy
values must also be taken with caution due to this metric's known overexaggeration of success in
cases of unbalanced classes (Chicco & Jurman, 2020; Tharwat, 2018). This is particulary important
to this case study since, for many reported subcatchments, the river channel accounts for much of
the subcatchment's area, thus unbalancing the classification matrix in favour of positive
observations. Thus, of the metrics reported herein, the Matthews Correlation Coefficient (MCC)
is considered to be the most representative of the success of the simulated floods – it is robust
against imbalanced classes while simultaneously requring high hit rates, low false alarms, high
correct rejections, and low miss rates to yield a high value.

Figure 11 visualizes the 100-year return period simulated flood for the Grand River watershed.
Although the colours of the simulated flood represents depth, the depth values have been excluded
as the sole focus of this test is extent. Inset maps are provided which highlight one subcatchment
with a high MCC (A, MCC= 0.95) and two subcatchments with low MCCs (B, MCC =0.34 and
0.38). The simulated flood shown in Figure 9A compares very well to the extent of its observed
counterpart, suggesting that the high MCC values do represent areas of strong model success.
Notably, three hydrometric stations are located within the Figure 11A subcatchment: 02GA014,



02GA027, and 02GA016. Per the methods in Section 2.2.2, station 02GA014 yielded a period of
record of 54, 02GA027 yielded an insufficient (<10) period of record, and station 02GA016
yielded a period of record of 58. The presence of the two hydrometric stations with a considerable
periods of record likely strengthened the regional regression of the area and contributed to the
success of the simulated flood shown in Figure 11A. In contrast, within the low-MCC (0.34 and
0.38) subcatchments shown in Figure 11B the simulation considerably overestimated the extent of
the 100-year return period flood. The overestimation of the flood extents observed in Figure 11B
can likely be attributed, at least in part, to the following. It was observed (Figure S5) that dams
(Grand River Conservation Authority, 2000) are located both upstream and downstream of the
area shown in Figure 11B. The current iteration of the model makes no provision for flood
mitigation structures. As such, the model has likely overestimated the discharge values at
subcatchment outlets, particularly for those outlets which are a) relatively downstream in the
watershed and b) impacted by nearby structures. However, it's possible to include such operations
in future versions of the model by either modifying the DEM values to reflect flood control
structures or by offsetting the discharge of the catchment based on structure storage.

With respect to the Ottawa River watershed, Figure 12 highlights subcatchments whose
comparison between observed and simulated flood extents yielded low (A: MCC= 0.16 ; B: MCC=
0.29), moderate (D: MCC =0.67) and high (C: MCC = 0.91) MCC values. As with Figure 11, the
colour of the simulated floods represnts depth, but depth values have been excluded as the sole
focus of the MCC test is on flood extent. Figure 12A shows the simulated and observed flood
extents for return period 25.69. Two main factors influencing the low MCC are readily apparent.
The first is that the observed FEP appears "cut off", not extending through most of the
subcatchment. It is possible that the flood in the remainder of the subcatchment was simply not
digitized during the observed FEP's generation, especially given the subcatchment's position.
However, of the area of the subcatchment intersected by the observed FEP, the simulated flood
has considerably underestimated the observed flood extent. Figure 12B shows the extent
comparison of the 42.69 -year return period in a subcatchment of low MCC (0.29). Interestingly,
the simulated flood was not as vastly different from the observed flood as the very low MCC value
might suggest, particulary with reference to Figure 11B, which yielded slightly higher (0.34 and
0.38) MCC values. The most visually prominent discrepancy in Figure 12B appears to be



connected to a false positive section near the south side of the subcatchment, which is consistent
with the subcatchment's moderately high False Positive Rate (0.41) and high False Discovery Rate
(0.84). Figure 12C illustrates a subcatchment of high MCC (0.91), characterized by an overall
underestimation in flood extent, barring a slight overestimation in one area. Figure 12D (MCC =
0.67) shows a mixture of overestimation and underestimation.

Table 4 lists the number of subcatchments evaluated, the minimum MCC, the median MCC, and
the maximum MCC for each of the 5 test return periods. The median MCC values ranged from
0.67 to 0.94, with both of those values coming from the Ottawa River watershed (return periods
42.69 and 26.5, respectively). The median MCC for the Grand River watershed was 0.84.
Additionally, the median $F_1$ score (Chicco & Jurman, 2020) for the Grand River watershed was
0.85. The median $F_1$ scores for Ottawa River watershed return periods 26.5, 16.52, 25.96, and
42.69 were 0.96, 0.87, 0.90, and 0.65 respectively. Such results are approximately in line with Lim
& Brandt (2019) which determined that low-resolution DEMs are capable of yielding relatively
high comparison metrics (eg $F_1$ values approximateely >= 0.80) in situations where Manning's n
varies widely over space. The connection between high values of Manning's n and flood
overestimation (false discovery) was also discussed. The Grand River watershed yielded a median
False Discovery Rate (FDR) of 0.20, and the four Ottawa River watershed cases yielded respective
median FDRs of 0.019, 0.01, 0.006, and 0.44 for the evaluated subcatchments. The moderately
high FDR value of 0.44 for the 42.69-year return period and the observed overestimation of flood
extent (Figure 12B) may be a result of high local Manning's n values. In addition, the influences
of flat terrain (Lim & Brandt, 2019)  and anabranch must be considered as it can disrupt the
assumption of a single drainiage direction for each pixel during subcatchment delineation. The
topography of the area of the Ottawa River watershed wherein the extent comparisons were made
is realtively flat with multiple anabranches and thus can lead to chaotic network delineation.
Although attempts were made in this model to counter this impact and avoid slope values of 0 (the
burning of the polyline network into the DEM, Section 2.2.1 and Figure 2), the use of the
Manning's equation was still compromised in certain areas and likely had a negative impact on
the resultant flood simulations.



Overall, the results indicated that the current iteration of the InundatEd flood model was reasonably
successful on the basis of moderate-high MCC values and direct comparisons. However, any
weight assigned to this claim must, in addition to the previously discussed caveats, recall that only
extent and not depth was compared between the observed and simulated floods. The use of the
DGGS big-data architecture provides a promising foundation for further work, such as the
incorporation of the impacts of flood control structures, on the InundatEd model.

### 3.4 Model Performance

Supplementary Figure S6 contrasts runtimes using the DGGS method against those using a
traditional, raster-based method for sub-catchments within the Grand River Watershed (n= 306 for
each method) during the generation of respective RP 100 flood maps. The mean runtime using the
DGGS method (0.23 seconds) was significantly lower than the mean runtime using the raster-
based method (3.98 seconds) at both the 99% confidence intervals ($p < 2.2e\text{-}16$). Thus, the
efficiency of the proposed inundation model -coupled with a big-data Discrete Global Grids
Systems architecture- is demonstrated with respect to processing times with limited input data. As
the IDEAS framework and the InundatEd flood modelling method continue to develop, processing
time benchmarks could be established to track and evaluate the model's robustness against
increasing complexity (e.g., the integration of hydrological processing algorithms) and to facilitate
comparisons with other inundation models.

### 3.5 Conclusions

We have tested a novel flood modelling and mapping system, implemented within a DGGS-based
big data platform. In many parts of the world, including Canada, the widespread deployment of
detailed hydrodynamic models has been hindered by complexities and expenses regarding input
data and computational resources, especially the dichotomy between processing time and model
complexity. This research proposes a novel solution to these challenges. First, we demonstrated
the development of a flood modelling framework in a Discrete Global Grid Systems (DGGS) data
model and the presentation of the models' outputs via an open-source R/Shiny interface robust
against algorithm modifications and improvements. The DGGS data model efficiently integrates
heterogeneous spatial data into a common framework, rapidly develops models, and can scale for
thousands of unit processing regions through easy parallelization. Second, the use of the





catchment-integrated Manning's equation avoids high-uncertainty river cross-sections and
produces physically justified flood inundation extents. Third, DGGS-powered analytics allow
users to quickly visualize flood extents and depths for regions of interest, with reasonable
alignment with observed flooding events. Finally, we believe our flood-inundation estimation
method can address situations where good quality data is scarce and/or there are insufficient
resources for a complex model. To apply the model in a real time environment we would need a
discharge forecasting model or have real-time discharge data at the catchment outlet, which could
be used to compute the flood inundation using the pre-computed stage-discharge relationship and
inundation model.





























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



**List of tables:**


Table 1. Values of Manning's n

| NRCAN LULC Value | NRCAN Description | Manning's n |
|---|---|---|
| 1 | Temperate or sub-polar needleleaf forest | 0.16 |
| 2 | Sub-polar taiga needleleaf forest | 0.16 |
| 5 | Temperate or sub-polar broadleaf deciduous forest | 0.16 |
| 6 | Mixed forest | 0.16 |
| 8 | Temperate or sub-polar shrubland | 0.1 |
| 10 | Temperate or sub-polar grassland | 0.035 |
| 12 | Sub-polar or polar grassland-lichen-moss | 0.035 |
| 13 | Sub-polar or polar barren-lichen-moss | 0.03 |
| 14 | Wetland | 0.1 |
| 15 | Cropland | 0.035 |
| 16 | Barren lands | 0.025 |
| 17 | Urban | 0.08 |
| 18 | Water | 0.04 |




Table 2. Study Watershed Characteristics

| Characteristic | Grand River Watershed | Ottawa River Watershed |
|---|---|---|
| Drainage Area (km$^2$) | 6,800 (Li et al., 2016) | 146,000 (Nix, 1987) |
| Elevation range (masl) | 173-535 (Lake Erie Source Protection Region Technical Team, 2008) | 430 – 20 (Nix, 1987) |
| Geologic characteristics | Underlain by groundwater-rich, fractured, porous limestone bedrock; surface geology characterized by glacial till and moraine complexes (Liel et al., 2016) | Incorporates the geological subdivisions St. Lawrence Lowlands, Grenville Province, Superior Province, and Cobalt Plate within the region of the Canadian Shield (Environment and Climate Change Canada, 2019) |
| Approximate Population size | 985,000 (Grand River Conservation Authority, 2014) | > 2,000,000 (Environment and Climate Change Canada, 2019) |
| Land Use / Land Cover | 43% agriculture; 26.92% range-grass and pasture; 12% forests; 9.29% urban areas; 1.8% wetlands (Veale & Cooke, 2017) | 73% forested (Quebec); 85% mixed and deciduous forest, 15% boreal (middle-south and northern regions, respectively) (Environment and Climate Change Canada, 2019); 6% farmland; <2% developed (Werstuck & Coulibaly, 2017) |
| Average Annual Precipitation (mm) | 800-900 (Kaur et al., 2019) | 840 (Werstuck & Coulibaly, 2017) |
| Temperature | 8-10 ° C average annual; moderate-to-cool temperate (Kaur et al., 2019) | 21 - -10 °C average daily (Werstuck & Coulibaly, 2017) |






Table 3. Simulated Flood Generation – Ottawa River Watershed

| Observed Flood Extent Polygon | Observed Date and Time (UTC) | Intersected Hydrometric Station | Station Period of Record (years) | Index Flood ($\tilde{Q}$, m³s⁻¹) | Observed Discharge (m³s⁻¹) | Logspline fit observation count | Cumulative Probability Value | Return Period (years) |
|---|---|---|---|---|---|---|---|---|
| FloodExtentPolygon_QC_ LowerOttawa_20190429_ 230713.shp | 2019/04/29 23:07:13 | 02KF005 | 38 | 3400 | 5790 | 1487 | 0.962 | 26.5 |
| FloodExtentPolygon_QC_ LowerOttawa_20190507_ 111329.shp | 2019/05/07 11:13:29 | 02KF005 | 38 | 3400 | 5350 | 1487 | 0.939 | 16.52 |
| FloodExtentPolygon_QC_ LowerOttawa_20190513_ 225800.shp | 2019/05/13 22:58:00 | 02KF005 | 38 | 3400 | 5570 | 1487 | 0.961 | 25.96 |
| FloodExtentPolygon_QC_ CentralOttawa_20190503_ 113004.shp | 2019/05/03 11:30:04 | 02KB001 | 52 | 258 | 477 | 1487 | 0.977 | 42.69 |






Table 4. Matthews Correlation Coefficient Results

| Watershed | Return Period (years) | Number of evaluated subcatchments | Minimum MCC | Median MCC | Maximum MCC |
|---|---|---|---|---|---|
| Grand River | 100 | 71 | 0.33 | 0.84 | 0.98 |
| Ottawa River | 26.5 | 17 | 0.49 | 0.94 | 1.00 |
| Ottawa River | 16.52 | 21 | 0.13 | 0.80 | 1.00 |
| Ottawa River | 25.96 | 22 | 0.16 | 0.85 | 1.00 |
| Ottawa River | 42.69 | 7 | 0.29 | 0.67 | 0.74 |




**List of Figures**
Figure 1. GIS Input Data – Grand River Watershed (a) and Ottawa River Watershed (b)
Topography. The maps are created in Qgis with the basemaps provided by © Google Satellite
Maps under OpenLayerPlugin.

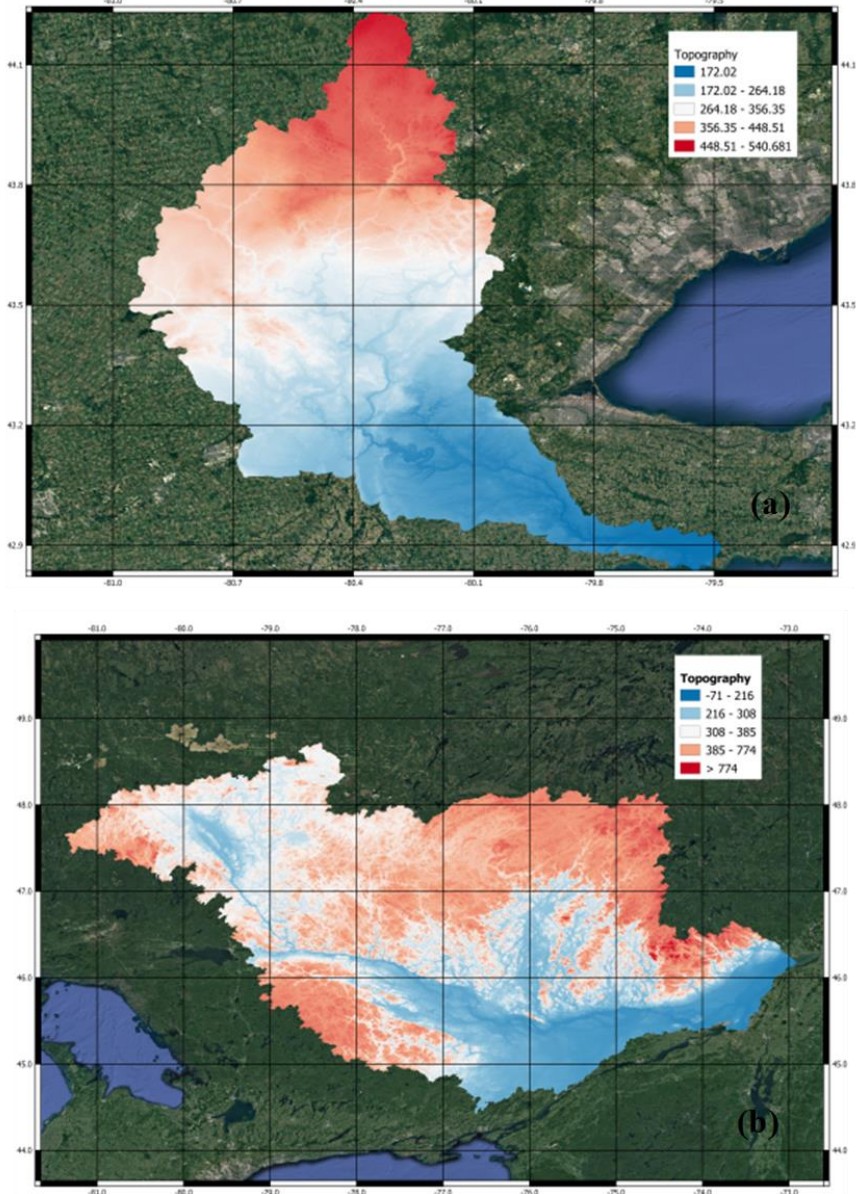




Figure 2. Flood Modelling Stage 1: GIS Preprocessing

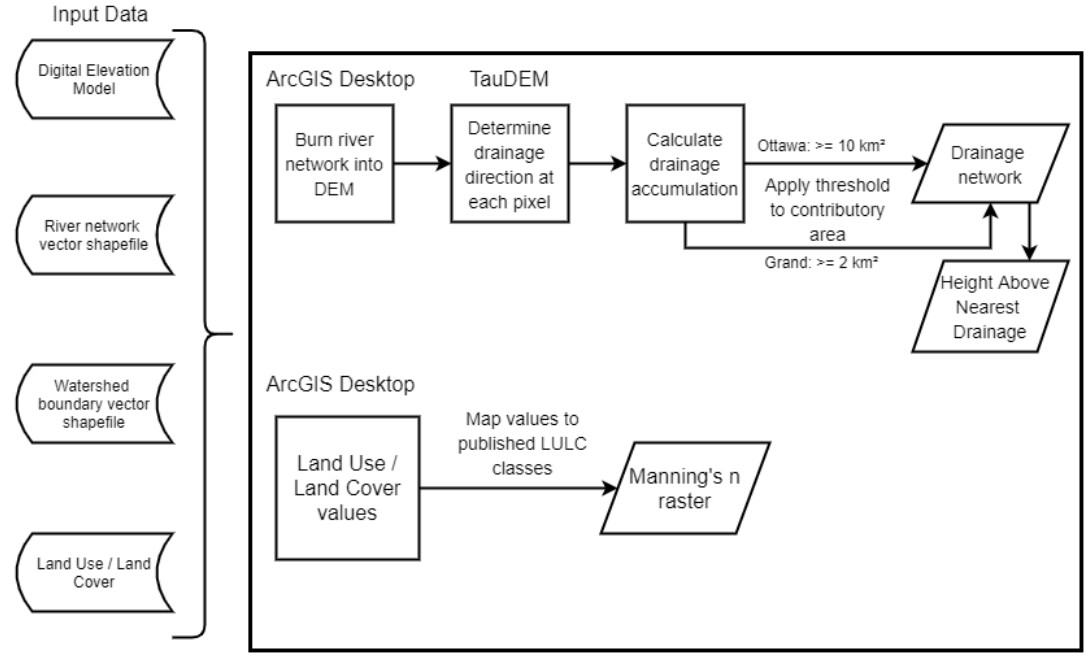




Figure 3. Flood Modelling Stage 2: Flood Frequency Analysis and Regional Regression

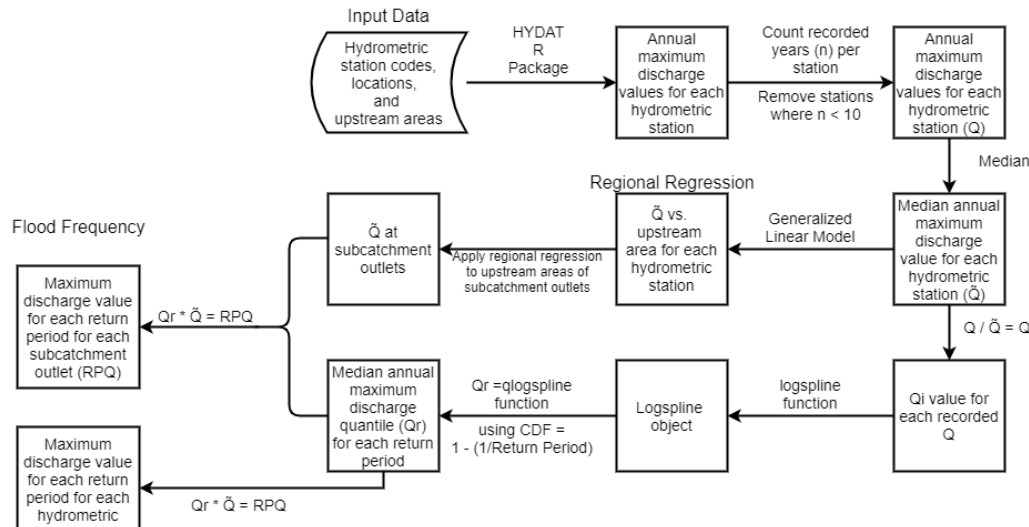




Figure 4. Flood Modelling Stage 3: Catchment Integrated Manning's Equation





Figure 5. InundatEd User Interface (a) and System Diagram (b). The basemap is created in Leaflet
using © OpenStreetMap contributors 2020. Distributed under a Creative Commons BY-SA
License

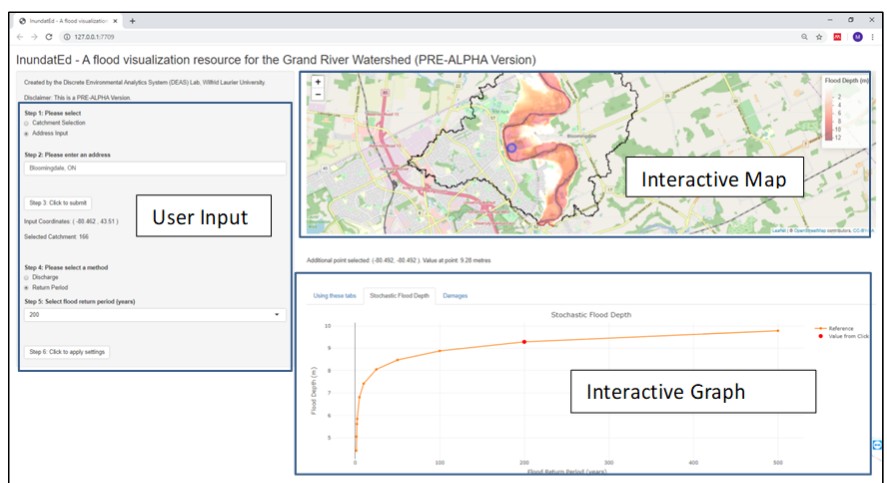

a)    InundatEd User Interface

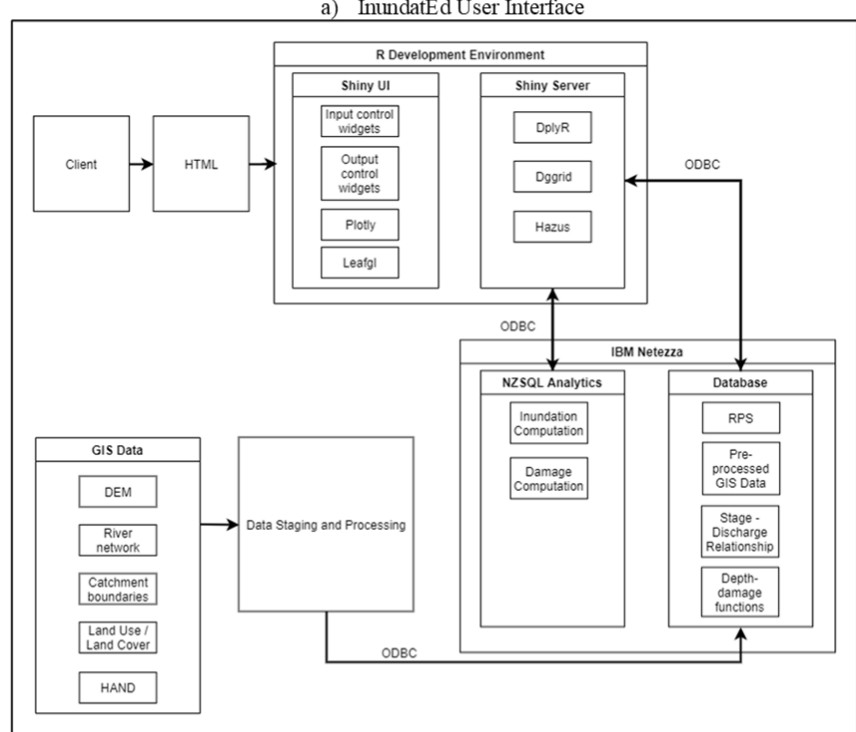

b)    InundatEd System Diagram





Figure 6. GIS processing outputs for the Grand River Watershed and the Ottawa River Watershed:
Height Above Nearest Drainage (a-b), Drainage network (c-d), and Manning's n values (e-f). The
maps are created in Qgis with the basemaps provided by © Google Satellite Maps and © Google
Street Maps under OpenLayerPlugin.

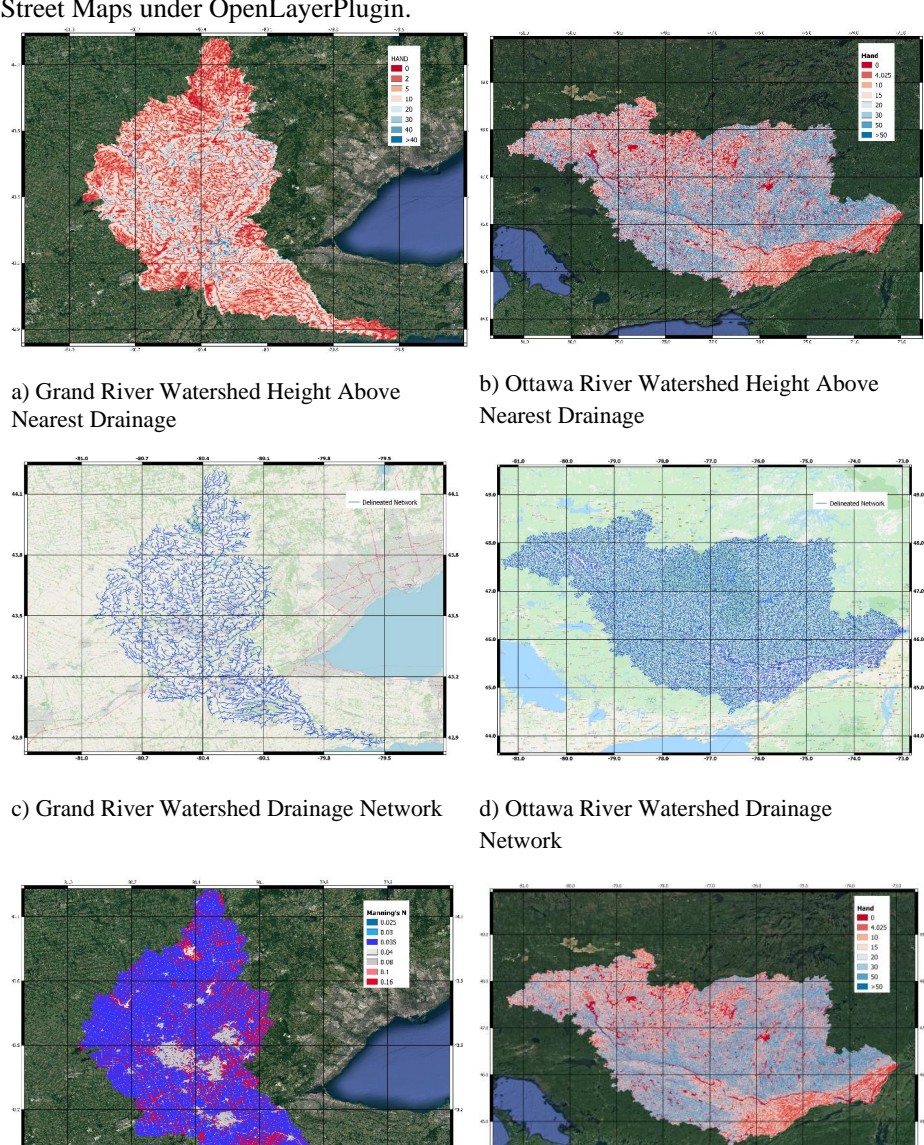

a) Grand River Watershed Height Above Nearest Drainage

b) Ottawa River Watershed Height Above Nearest Drainage

c) Grand River Watershed Drainage Network

d) Ottawa River Watershed Drainage Network

e) Grand River Watershed Manning's n

f) Ottawa River Watershed Manning's n




Figure 7. Flood frequency and regional regression plots

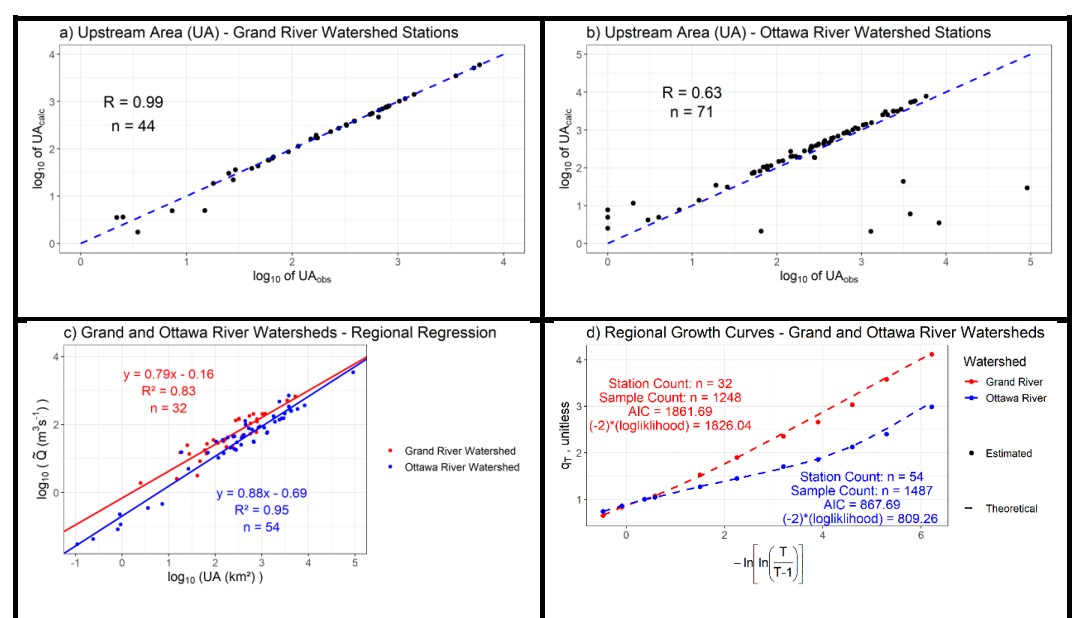






Figure 8. Theoretical Versus Estimated Flood Quantiles

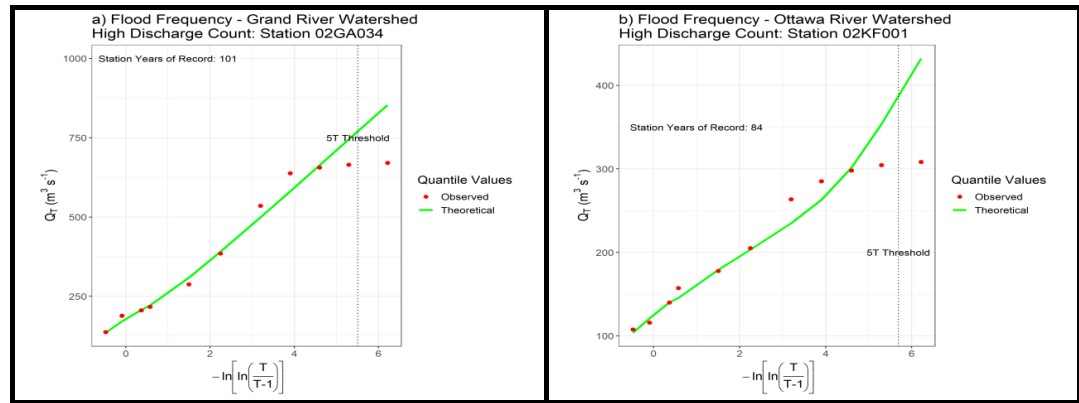





















Figure 9. Binary Classification Results – Grand River Watershed












Figure 10. Binary Classification Results – Ottawa River Watershed










Figure 11. Simulated Flood and Insets – Grand River Watershed 100-Year Return Period

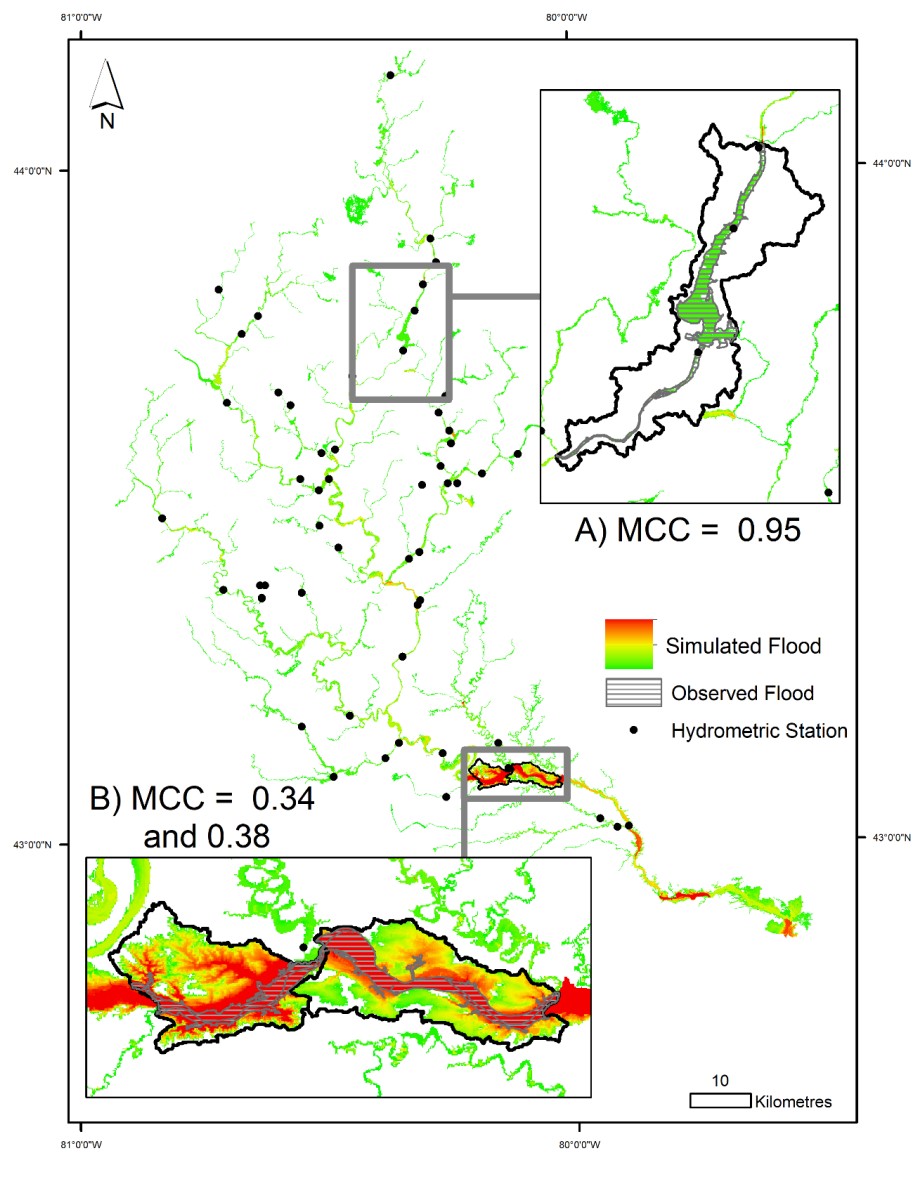




Figure 12. Observed and Simulated Flood Extents– Ottawa River Watershed



**Acknowledgement:**
Thank you, Majid Hojati and Amit Kumar, for assistance in GIS and software set up.
The flood extent products are derived from satellite images and ancillary data with a system
developed and operated by the Strategic Policy and Innovation Sector of Natural Resources
Canada © Department of Natural Resources Canada. All rights reserved.
Data credited to the Grand River Conservation Authority contains information made available
under Grand River Conservation Authority's Open Data Licence v2.0.





**Funding**

This work was funded by the Global Water Futures research programme under the Developing
Big Data and Decision Support Systems theme.

**Conflicts of interest/Competing interests**

The authors declare that there are no competing interests.

**Availability of data and material**

Any data that support the findings of this study, not already publicly available, are available from
the corresponding author, C. Chaudhuri, upon reasonable request.

**Author Contribution**

The idea behind this research was conceived, implemented, and written equally by all the authors.

**Code availability**

The current version of InnundatEd is available from the project GitHub
website: https://github.com/thespatiallabatLaurier/floodapp_public. The exact version of the
model used to produce the results used in this paper is archived on Zenodo
(*10.5281/zenodo.4095618*).