# Peer review of "InundatEd: A Large-scale Flood Risk Modeling System on a Big-data - Discrete Global Grid System Framework"

_Geoscientific Model Development, 2020_

## Referee Comment (RC1) · Anonymous Referee #1 · 12 Nov 2020

Comments on

"InundatEd: A Large-scale Flood Risk Modeling System on a Big-data - 1 Discrete Global Grid System Framework"

Chiranjib Chaudhuri, Annie Gray, and Colin Robertson

The paper presents a simple flood modelling framework model based on HAND to predict flood levels in two watersheds in Ontario and Quebec using a big-data discrete global grid systems-based architecture with a web-GIS platform. The authors indicate that the combination of simple conceptual flood method with big-data approaches remains largely uninvestigated, but they don't make a clear demonstration of what their big-data processing system brings that cannot be accomplished by existing large-scale flood modelling methods at the continental scale.

There is little mention of uncertainty concerning flood estimates in this study, and some of the discrepancies between theoretical and estimated values (e.g. Figure 7b; Figure 8 for large return periods) are dismissed without enough analysis on the implications on the predicted flood zones. The justification for some methodological steps needs to be improved, for example the Lotter method, which is used here despite being singled out in the cited reference (Tullis, 2012) as the only approach "not recommended for use".

The paper is very long, with several figures, and could be better synthesized to focus on the novelties brought by this study, since there are many other large-scale flood modelling approaches now available. I made suggestions to remove some figures below. I believe that a shorter version, which would include the CSI to better compare with other large-scale flood modelling approaches, could be acceptable for publication once the comments identified below have been addressed.

**Detailed comments**

Line 71: The list of references for simpler models cited here should include large-scale flood modelling approach such as LISFLOOD-FP that have been used successfully to produce flood maps at the continental scale (e.g. Wing al. 2017). Also, one of the cited references (Oubennaceur et al. 2019) state on p. 46 that "Inundation maps of the Richelieu River were derived with the 2D simulator H2D2" (where H2D2 is a 2D hydrodynamic model). If I understand well their approach, they used this model's results to develop a simple power function relating discharge to water surface elevation, but it is not clear how they could have obtained this relationship without the H2D2 model. Therefore, can this study be considered a "simple conceptual model"?

Line 171: What is the resolution of the DEM, and what is the vertical accuracy? The LULC data should also be described in more detail (resolution, accuracy). The drainage area of both

watersheds (Grand River and Ottawa River) should also be provided here (this information is given in Table 2 which is only presented on line 377).

Line 174: It would be useful to add the (32 and 54) gauging stations used in the analysis for both watersheds on Figure 1. Why is the legend for topography for the Ottawa River starting at a negative value (-71 m)?

Line 198: Is the density of gauging station in Canada comparable to that in Norway in the study of Hailegeorgis and Alfredsen (2017), and does this make a difference in our confidence in a regional approach in Canada? The reference should be Dalrymple (instead of Darlymple).

Lines 215-216: "Only stations with a period of record >= 10 years of annual maximum discharge were maintained (n = 32 and n = 54, respectively)." A minimum of 10 years of annual maximum discharge values seems very low (the minimum in Hailegeorgis and Alfredsen, 2017 was 22 years). It would also be useful to add "for the Grand River and Ottawa River watersheds" before "respectively" as it is not obvious in this sentence.

Line 217: Providing the median or average period of records for both watersheds would be useful.

Lines 246-247 : " for use in a watershed where the flow has been modelled due to human abstraction is a fundamental step of the analysis process and must account for disturbance-related changes to the extreme value characteristics of the flow". It is not clear what you mean by "modelled due to human abstraction" in the context of the two studies watersheds, or what disturbance-related changes are expected, so providing more information here would be useful (some of that information is presented later in Table 2). The Ottawa River, for example, is a very large watershed, with its upstream parts mainly forested, so you need to clarify what disturbances have affected its hydrology since in Table 2 you indicate only 6% farmland and < 2% developed.

Lines 249-250: Here again, it would help to clarify what artificial abstraction you are referring to and how this is supposed to affect extreme value characteristics of the flow. If we look at the causes of the major floods of 2017 and 2019 in the Ottawa watersheds, they look mainly natural (very snowy winter followed by very wet spring, with deeply frozen soil due to very cold temperatures in the autumn, thus limiting infiltration).

Line 259 : " Per assumption c of the index flood method…". More information is needed to understand what assumption c entails. A reference would also be useful.

Line 262 : Qi needs to be defined and Qtilde should be more clearly defined as the median annual maximum discharge.

Lines 298-299: Note that other large-scale flood modelling approaches also don't require channel geometry (e.g. Wing et al. 2017)

Lines 311-312: Do you have an uncertainty estimate on slope estimated with the 30m x 30m DEM? As indicated above, having more information on the vertical accuracy of the DEM would be useful.

Line 320: Why did you choose the Lotter method? In Tullis (2012), it is stated (p. 72) that "Pillai (1962) concluded that the Horton relationship performed the best and that the Lotter relationship gave inconsistent results." On the same page, Tullis (2012) indicates: "Flintham and Carling (1992) evaluated the Horton, Colebatch, Pavlovskii, and Lotter methods. They concluded that the Pavlovskii relationship was the most accurate, the Horton and Colebatch relationships were satisfactory, and the Lotter relationship performed poorly. Four of the five relationships evaluated in the three different studies were identified at least once as a "best performer," but consensus was not achieved regarding an overall best method. The Lotter relationship, on the other hand, was singled out in each study as "not recommended for use."

Line 325: (Figure 4) This figure has a lot of details which may not be needed, particularly since the HAND method is widely known.

Line 397: These 2 stations should be identified on Figure 1

Line 403: divided (instead of divded)

Line 447: Are there references related to flood modelling validation for the use of Matthews Correlation Coefficient? The reference cited here (Chicco & Jurman, 2020) is in the Genomics field. The Critical Success Index (CSI) seems more commonly used for flood modelling validation (e.g. the two listed references), so why not use it here to facilitate comparisons with other studies, for example Wing et al. (2017) who obtained a score of 55.2% for their flood maps of the United States.

Line 456 (Figure 6): I don't think this figure is needed. Providing HAND results (what are the units on Figure 6a,b?) at the scale of such watersheds is not really useful, and there is also little value to showing the drainage network or the Manning's n values, already presented in a table.

Lines 464-466: "The difference in correlation quality can be accounted for in part by the difference in the relative complexities of the delineated networks of the Grand River and Ottawa River watersheds." This is not a sufficient explanation for such a marked difference in correlation values between the two watersheds. What are the "relative complexities" of the network of the Ottawa River watershed that would explain that several points are not at all following the 1:1 slope?

Line 482: The information on dams on the Ottawa and Grand River watersheds should be provided here. To the best of my knowledge, about 40% of the flow is controlled by dams. What is the situation on the Grand River? As indicated below, dam information presented in supplementary material could easily be integrated into Figure 1.

Lines 490-491: "As expected, for the stations with high observation counts (n = 101 and n = 84 for the Grand River watershed (Figure 8a) and Ottawa River watershed (Figure 8b), respectively) the theoretical and estimated return periods are closer, at least for lower return periods." I find this sentence confusing. First, it seems to imply that there are stations with lower observation counts, but these are the only two stations presented. Then, the theoretical and estimated return periods are indeed close for low return period, but not at all for longer return period (even when less than the 5T threshold), which seems problematic.

Lines 497-505: The link between this paragraph and the previous ones is not clear and, overall, this paragraph seems out of place. It is the first time that the hexagonal gridding system is mentioned, and it is difficult to understand why it is problematic. I suggest removing this paragraph.

Lines 518: Why did you not include the CSI, since you are testing 4 metrics?

Line 520: (Figure 10): Why is the scale not the same in each of the figures? And why is the area covered for RP 42.69 different from the other maps? Table 4 should be mentioned here, as it is easier to compare with other large-scale flood modelling approaches (e.g. Wing et al. 2017) with actual values than with maps. It would also be easier to use the CSI for this comparison.

Line 527-528 : Wing et al. (2017) also used a 30-m DEM, and Sampson et al. (2015) a 90-m DEM, and they obtained a fit index of 55.2% and 75%, respectively. Perhaps the problem is more related to the HAND approach compared to the hydraulic modelling approach?

Line 531: Considering that you are not providing a lot of information on the uncertainty in, for example, the slope estimates (see above comment), it remains difficult to be convinced that the problem is with the flood extent polygons. It is also interesting to note that Lim and Brandt (2019), cited here, list CSI in their approaches, but not MCC. Further justification is needed for not using CSI in this study.

Line 545: Again, it is difficult to understand why an index that would allow for comparison with previous studies (such as the CSI) was not used. Providing references where MCC was used to assess the success of simulated floods would be useful.

Line 550: The depth value should still be indicated in the green-red colour scheme on Figure 11. Otherwise, you should use a single fill colour (the same comment applies to Figure 12).

Line 554: It seems obvious that a MCC of 0.95 is associated with a strong model success, so I am not sure to understand what is meant here.

Line 563: The position of dams should be indicated in Figure 1, not in supplementary material.

Line 568: It is possible (instead of it's possible)

Line 576: So a single fill colour should be used instead of the green-red legend in Figure 12.

Line 593: This should have been mentioned earlier, when presenting Figures 9 and 10.

Lines 605-607: "The moderately high FDR value of 0.44 for the 42.69-year return period and the observed overestimation of flood extent (Figure 12B) may be a result of high local Manning's n values." It is not clear why high Manning's n values would only play a role in this case.

Line 611: relatively (instead of realtively)

Line 613: It is the first time the burning of the polygon network is mentioned.

Supplementary material: Table S1: The number of digits after the decimal point should be consistent and reasonable (e.g. 5 digits for discharge values in $m^3$/s is too many). The same comment applies to Table S2.

Line 724: Dalrymple (instead of Darlymple)

---

## Short Comment (SC1) · 14 Nov 2020

Dear authors,

in my role as Executive editor of GMD, I would like to bring to your attention our Editorial version 1.2:

https://www.geosci-model-dev.net/12/2215/2019/

This highlights some requirements of papers published in GMD, which is also available on the GMD website in the 'Manuscript Types' section:

http://www.geoscientific-model-development.net/submission/manuscript_types.html

In particular, please note that for your paper, the following requirement has not been met in the Discussions paper:

- "The main paper must give the model name and version number (or other unique identifier) in the title."

Please add a version number for InundatEd in the title upon your revised submission to GMD.

Yours,

Astrid Kerkweg

---

## Referee Comment (RC2) · Anonymous Referee #2 · 22 Dec 2020

The authors present a detailed paper on the coupling of Regional Flood Frequency Analysis (RFFA) to a simple Manning's-based Height Above Nearest Drainage (HAND) conceptual model. While the efforts of the authors with respect to computer science are likely commendable, there are a number of issues with the hydrologic science that should preclude publication in its present form. I sincerely hope the below feedback is a valuable tool in the reformulation of this analysis and its write-up.

1. The paper contains much extraneous detail and a number of unnecessary figures, creating a long paper that is difficult to follow in places. Consider which information the reader requires to understand your model, how it works, and how it performs. For

instance, equations 2 and 8, figure 6, parts of section 2.2.2 and 3.1.

2. The novel aspects of this framework either do not exist or are inadequately emphasised. The presented RFFA does not appear to be much different to Hailegeorgis & Alfredsen (2017). Much of the Canadian RFFA literature by the likes of Taha Ouarda and Donald Burn is omitted. Advances in large-scale RFFA have been presented in, for instance, Faulkner et al. (2016, doi:10.1080/07011784.2016.1141665) or Smith et al. (2015, doi:10.1002/2014WR015814) and so the authors should be clear about what is novel about their approach. Similarly, the use of HAND in flood inundation prediction is well documented and so the authors must make clearer what is novel about their approach. Again, key literature on this front such as Afshari et al. (2018, doi:10.1016/j.jhydrol.2017.11.036), Liu et al. (2018, doi:10.1111/1752-1688.12660), and Zheng et al. (2018, doi:10.1111/1752-1688.12661) is missing.

3. The limitations on the functionality of the presented model are inadequately discussed. How does the requirement for quality river gauge data with long records impair the ability to deploy this model at large scales elsewhere? The limitations and inaccuracies of 'planar' models such as HAND are well known, but this is not discussed to any meaningful degree. The suggested literature above, amongst others, shows how physics-lite modelling approaches often correspond poorly with observations of flood inundation.

4. The inferences made arising from model validation results are often unsupported. For instance, the wild overprediction in Figure 11b is not unusual for models not grounded in a derivation of the shallow water equations. Where the benchmark flood is not valley filling and takes place in a wide, flat floodplains – as seems the case in this panel – the failure to simulate the flow of water can often lead to overprediction. Instead, the authors suggest grid resolution may be the issue. I suggest a more in-depth analysis in this section with evidence for the conclusions drawn.

5. The lack of a requirement for channel geometry is not clear to me. An understanding

of how much flow remains in-channel, which would have no meaningful representation in the DEM, would surely create a much more accurate model. Indeed, I do not know how one can hope to simulate floods such as 1.25, 1.5, 2.0, 2.33 year recurrence (most of which would presumably remain in-bank) without understanding channel conveyance. I think this needs to be further unpacked.

6. Damage computation is mentioned, but not demonstrated or tested. Consider dropping this component or illustrating a use case – as presently there is no scientific contribution on this front.

7. Validation results require much further explanation and contextualisation (grounding in literature). For instance, I have no idea what to take from lines 472-477. The Matthews Correlation Coefficient is not widely used and so makes contextualisation impossible. The reader needs more help in understanding what the validation results mean. Equally, it is unclear whether the authors have made errors in their computation or whether the MCC is unfit for purpose: any metric which rewards figure 11b higher than 12b is clearly not doing its job. Consider a more widely used metric so the reader can understand, to some extent, how your model compares to others in this field. Secondly, consider the purpose of the validation in the context of the purpose of the model. What is the point of your model? What is it meant for? How good does it therefore need to be? If you are computing damages, your benchmark may be higher as this requires accurate depths – if so, test how well the model replicates depths.

8. Some of the (necessary) figures require improvement. The colour scales on figures 9 and 10 make it difficult to discriminate 'good' from 'bad'. There is no key on the depth grid for figures 11 and 12, but should just be made a single block colour anyway as this is a binary comparison.
* * *

---

## Author Comment (AC1) · 22 Jan 2021

Reply to reviewers:

SC1: "The main paper must give the model name and version number (or other unique identifier) in the title."

The title of the paper has been modified to "InundatEd-v1.0: A Large-scale Flood Risk Modeling System on a Big-data - Discrete Global Grid System Framework"

RC1:

1. The paper presents a simple flood modelling framework model based on HAND to predict flood levels in two watersheds in Ontario and Quebec using a big-data discrete global grid systems-based architecture with a web-GIS platform. The authors indicate that the combination of simple conceptual flood method with big-data approaches remains largely uninvestigated, but they don't make a clear demonstration of what their big-data processing system brings that cannot be accomplished by existing large-scale flood modelling methods at the continental scale.

Our discrete global grid-based flood modeling approach addresses multiple computation- related concerns that often arise in large-scale flood modeling systems. These solutions and developments, listed below, have been incorporated into the manuscript's introduction and highlight the novel aspects of our approach and its application. The relevant text, are copied below:

"A recently developed DGGS-based data model and modelling environment is one such system which implements a multi-resolution hexagon tiling data structure within a hybrid relational database environment (Robertson, Chaudhuri, Hojati, & Roberts, 2020). Notably, and in contrast to previous systems, the only special installation entailed by the DGGS-based data spatial model is a relational database. As such, DGGS-based data model is agnostic and can be ported to any software-hardware architecture as long as it supports a relational database system  The system exploits the hardware capability of the database itself which can potentially incorporate the following: GPU(s), distributed storage, and a cloud database. In this paper we employ the IDEAS framework for the efficient computation, simulation, analysis, and mapping of flood events for risk mitigation in a Canadian context. As such, the novelty of this study is threefold: 1) the presentation of the new DGGS-based data model , 2) the contribution of this big-data approach to the field of flood modelling, and 3) the presentation of a web-interface which lets user compute the inundation on the fly based on input discharge for select Canadian regions where flood risk maps are either not publicly available or do not exist. Moreover, the properties and structure of the DGGS-base spatial data model address a number of challenges and limitations faced by previous flood modelling approaches in the literature. For instance, it is modular, making it easy to switch between RFFA-based, HAND-based, or alternative models without sacrificing the consistency of the framework. Likewise, the method by which Manning's n is calculated can be easily interchanged. Another novel aspect of this framework is the incorporation of Land Use Land Cover data in the estimation of the roughness coefficient Manning's n. This is in contrast to the literature,

wherein a constant value or a channel-specific value of Manning's n is typically used (Afshari et al., 2017; Zheng et al., 2018). In terms of the tradeoff between model complexity and computation power as discussed above, the IDEAS framework uses an integer-based addressing system which makes it orders of magnitude more efficient than that of other, more traditional spatial data models. This, in turn, benefits any and all spatial computations associated with flood modelling. Finally, whereas most major spatial computations entail specialized software/code, in the DGGS-based method the spatial relationship is embedded in the spatial-data model itself. Thus, the spatial relationships need not be considered beyond the use of certain rules of the spatial-data model. "

2.   There is little mention of uncertainty concerning flood estimates in this study, and some of the discrepancies between theoretical and estimated values (e.g. Figure 7b; Figure 8 for large return periods) are dismissed without enough analysis on the implications on the predicted flood zones. The justification for some methodological steps needs to be improved, for example, the Lotter method, which is used here despite being singled out in the cited reference (Tullis, 2012) as the only approach "not recommended for use".

The discrepancy in what is now Figure 4b arose primarily from the limited resolution (number of decimal places in lat-long) of the station location information; incorrect reporting of station locations and/or their drainage area (Environment Canada reported the drainage area as 0 for multiple stations); and sometimes wrongly snapping stations to the tributaries rather than to the main river, particularly in cases involving a wide river channel or braided river. For example, station 02LA012 (a, below) intersects one of the drainage tributaries in the model, but actually it is a part of the main river. Station 02KE002 (b, below) is almost a kilometre away from the main river by location. We earlier employed a 300m search radius to search for the nearest river for every station, but in the borderline cases this method failed. However, this does not affect the model itself, as we have used the station-specific drainage areas reported by Environment Canada to create the regional regression model. Section 3.1 has been edited to include this brief discussion. Additionally, we have copied below (c) a version of Figure 4b which resulted from manually selecting the channel for each station. The strong positive correlation (0.99) indicates that the discrepancy was location-related, as opposed to a shortcoming of our network delineation process.

[Figure]

c) Figure 4b using manual snapping:

[Figure]

b) Upstream Area (UA) - Ottawa River Watershed Stations

With respect to the theoretical vs estimated quantiles, the assumption of homogeneity for the entire watershed is one of the major sources of disagreement. Estimations of higher return periods, considering the 5T rule, would require more observations. However, further sub-sampling the stations into regional homogeneous groups would have reduced the data quantity significantly for each group. Therefore, we decided to do it for the entire catchment. Unfortunately, this substantially affected the higher quantiles of the estimated discharge. We have added a discussion to this effect in Section 3.1. It is copied below:

"However, and as shown in Figure 5, the theoretical and estimated quantiles deviate at lower RP values than the 5T threshold for both stations. This disagreement between the theoretical and estimated quantiles recalls the assumption of homogeneity for each watershed (Burn, 1997) - estimations of higher return periods, considering the 5T rule, would require more observations. However, further sub-sampling the stations into regional homogeneous groups would have reduced the data quantity substantially for each group."

3. The paper is very long, with several figures, and could be better synthesized to focus on the novelties brought by this study, since there are many other large-scale flood modelling approaches now available. I made suggestions to remove some figures below. I believe that a shorter version, which would include the CSI to better compare with other large-scale flood modelling approaches, could be acceptable for publication once the comments identified below have been addressed.

We have substantially reduced the number of figures and included the CSI comparison.

4. Line 71: The list of references for simpler models cited here should include large-scale flood modelling approach such as LISFLOOD-FP that have been used successfully to produce flood maps at the continental scale (e.g. Wing al. 2017). Also, one of the cited references (Oubennaceur et al. 2019) state on p. 46 that "Inundation maps of the Richelieu River were derived with the 2D simulator H2D2" (where H2D2 is a 2D hydrodynamic

model). If I understand well their approach, they used this model's results to develop a simple power function relating discharge to water surface elevation, but it is not clear how they could have obtained this relationship without the H2D2 model. Therefore, can this study be considered a "simple conceptual model"?

We agree the use of H2D2 when estimating the parameters of the model should be considered when calling the model "simple conceptual". We changed the terminology accordingly and the edited text is copied below:

"A class of model which uses the output of a more complex model as a means of calibrating a relatively simpler model is also gaining popularity (Oubennaceur et al., 2019)."

5. Line 171: What is the resolution of the DEM, and what is the vertical accuracy? The LULC data should also be described in more detail (resolution, accuracy). The drainage area of both watersheds (Grand River and Ottawa River) should also be provided here (this information is given in Table 2 which is only presented on line 377).

We have revised the introduction of the input data to include the resolutions of the DEM and LULC data (both 30m x 30m, respectively) and the vertical accuracy of the DEM (0.34 m ± 6.22 m, i.e., 10 m at 90 percent confidence level). The drainage areas of the Grand River watershed (6,800 square kilometers (Li et al., 2016) and the Ottawa River watershed (146,000 square kilometres, Nix (1987)) have also been added. The edited text and full references for the citations in this answer are copied below.

"The following GIS input data were obtained from Natural Resources Canada for the Grand River and Ottawa River watersheds and cropped to their respective drainage areas of 6,800 square kilometres and 146,000 square kilometers: Digital Elevation Models (Canada Centre for Mapping and Earth Observation, 2015); river network vector shapefiles (Strategic Policy and Innovation Centre, 2019); and Land Use Land Cover (LULC) (Canada Centre for Remote Sensing, 2019). Figure 1 shows the input Digital Elevation Model with elevation values given in metres, and the dams and gauging stations used in this study. The resolution of the DEM and LULC data is 30m x 30m. The vertical accuracy of the DEM is 0.34 m ± 6.22 m, i.e., 10 m at the 90% confidence level. The vertical datum used is the Canadian Geodetic Vertical Datum of 2013 (CGVD2013). The stations used for station-level discharge comparison are labeled in Figure 1."

Li, Z., Huang, G., Wang, X., Han, J., Fan, Y. (2016). Impacts of future climate change on river discharge based on hydrological interference: a case study of the Grand River Watershed in Ontario, Canada. *Science of the Total Environment, 548-549, 198-210.*

Nix, G. A. (1987). Management of the Ottawa River Basin. *Water International, 12(4), 183-188.*

6. Line 174: It would be useful to add the (32 and 54) gauging stations used in the analysis for both watersheds on Figure 1. Why is the legend for topography for the Ottawa River starting at a negative value (-71 m)?

Figure 1 is modified accordingly and is copied below. Additionally, the legends have been changed to a continuous scale. The Ottawa River watershed has a couple of quarries (eg. 45.640691, -73.545042) which have elevation of below the datum. We pitfilled those for hydrology processing. However, we have plotted the raw DEM here.

7. Line 198: Is the density of gauging station in Canada comparable to that in Norway in the study of Hailegeorgis and Alfredsen (2017), and does this make a difference in our confidence in a regional approach in Canada? The reference should be Dalrymple (instead of Darlymple).

In the interest of clarity, we consider gauge density in terms of the method used by WMO (2008) (full citation below) as SGD=DA/N where SGD is stream gauge density, DA is watershed drainage and N is the number of gauges.

WMO (World Meteorological Organization). 2008. *Guide to Hydrological Practices*, vol I, 6th edn. Geneva: World Meteorological Organization. WMO 168.

We can't comment on the difference in gauging density of networks between Hailegeorgis and Alfredsen (2017) and our studies. This is because Hailegeorgis and Alfredsen (2017) created multiple catchments with outlets to individual stations as part of their modelling process, whereas we have modelled each study watershed (Grand River and Ottawa River) in its entirety using multiple stations. Overall, there are many differences between the two contexts and methodologies which makes any attempt at 1:1 comparison redundant in terms of station density. However, we are able to compare in terms of the average station upstream area.

Hailegeorgis and Alfredsen (2017) used an average station upstream area of 449.03 sq km. Our methodology yielded an average upstream area of 581.17 sq km for the Grand River watershed and 3015.4 sq km for the Ottawa River watershed. The SGD values are 212.5 sq km/gauge in the Grand River watershed and 2703.7 sq km/gauge for the Ottawa River watershed.

The Dalrymple reference has been corrected in the manuscript and references list.

8. Lines 215-216: "Only stations with a period of record >= 10 years of annual maximum discharge were maintained (n = 32 and n = 54, respectively)." A minimum of 10 years of annual maximum discharge values seems very low (the minimum in Hailegeorgis and Alfredsen, 2017 was 22 years). It would also be useful to add "for the Grand River and Ottawa River watersheds" before "respectively" as it is not obvious in this sentence.

The threshold of a minimum of 10 years of annual maximum discharge values was selected based on England et al. (2018) (full citation below). This threshold is also employed in Faulkner, Warren, & Burn (2016) (full citation below). While we acknowledge that increasing the threshold, for instance to 22 years, would increase the data quality, this would also decrease the quantity of data available. For instance, a threshold of 10 years yields 1248 values of annual maximum discharge in the Grand River watershed, whereas a threshold of 22 years would reduce this to 1084. For the Ottawa River watershed, the quantity of data would drop from 1487 to 1068 records. This reduction, in turn, impacts the return periods which can be reasonably estimated, per the 5T rule. As discussed in Section 3.1, the 5T rule states that, to estimate the quantiles of a given return period T, a minimum of 5T years of annual maximum discharge data are required. Using the numbers from the Grand River watershed, the estimable return period would drop from approximately 248 years to approximately 216 years. Based on these considerations and the previous uses of a 10 year minimum in the literature, we deemed this to be an appropriate trade-off between data quality and data quantity. Of course, and as with other aspects of the modelling process, the InundatEd framework can easily incorporate methodological changes, including an increase in the minimum numbers of years of annual maximum discharge.

The order of the watersheds has been specified as requested. The edited text is copied below:

> "Only stations with a period of record >= 10 years of annual maximum discharge were maintained (n = 32 and n = 54 respectively for the Grand River watershed and the Ottawa River watershed) (England et al. (2018); Faulkner, Warren, & Burn (2016))."

England, J.F., Jr., Cohn, T.A., Faber, B.A., Stedinger, J.R., Thomas, W.O., Jr., Veilleux, A.G., Kiang, J.E., & Mason, R.R., Jr. (2018). Guidelines for determining flood flow frequency - Bulletin 17C (ver. 1.1). U. S. Geological Survey Techniques and Methods, book 4, chap. B5, 148 p.https://doi.org/10.3133/tm4B5.

Faulkner, D., Warren, S., & Burn, D. (2016). Design floods for all of Canada. *Canadian Water Resources Journal, 41(3)*, 398-411. 10.1080/07011784.2016.1141665.

9. Line 217: Providing the median or average period of records for both watersheds would be useful.

We have added the median periods of record for both watersheds. For the Grand River watershed, the median period of record is 50 years. For the Ottawa River watershed, the median period of record is 36 years. The updated text is copied below:

"The minimum, median, and maximum periods of record for the Grand River watershed were 12, 50, and 86 years, respectively. Periods of record for the Ottawa River watershed ranged from a minimum of 10 years to a maximum of 58 years with a median of 36 years."

10. Lines 246-247 : " for use in a watershed where the flow has been modelled due to human abstraction is a fundamental step of the analysis process and must account for disturbance-related changes to the extreme value characteristics of the flow". It is not clear what you mean by "modelled due to human abstraction" in the context of the two studies watersheds, or what disturbance-related changes are expected, so providing more information here would be useful (some of that information is presented later in Table 2). The Ottawa River, for example, is a very large watershed, with its upstream parts mainly forested, so you need to clarify what disturbances have affected its hydrology since in Table 2 you indicate only 6% farmland and < 2% developed.

The flow modification due to flow abstractions are listed below;

The Grand River watershed includes 11 overflow weirs (catchment area 36 - 5499 sq km); 8 flow control and flow augmentation structures (catchment area 2.7 - 6480 sq km); and 13 small dams (catchment area 1.8 - 2052 sq km).

For the Ottawa River watershed, solely dam information was available. A total of 26 dams are situated here, with the catchment area ranging from 7.3 sq km to 147,401 sq km.

These dams and flow augmentations have altered the flow significantly. The dams can effectively reduce the downstream flow in the downstream while raising the upstream water level. In terms of flood inundation, this can have a significant impact.

A number of human impacts have been added to this sanction to clarify our meaning. The updated text are copied below:

"The selection of a suitable probability distribution model – a common tool in hydrologic modelling studies (Langat et al., 2019; Singh, 2015)-for use in a watershed where the flow has been modified due to human impact – whether via development of built up areas, agriculture, road building, resource extraction activities such as forestry and mining, or flow abstraction in terms of dams and weirs is a fundamental step of the analysis process and must account for disturbance-related changes to the extreme value characteristics of the flow. Sometimes, natural hydrologic peaks, such as the spring freshet, are exacerbated by antecedent conditions such as large snowpacks and frozen soils, resulting in substantial flood events."

11. Lines 249-250: Here again, it would help to clarify what artificial abstraction you are referring to and how this is supposed to affect extreme value characteristics of the flow. If we look at the causes of the major floods of 2017 and 2019 in the Ottawa watersheds, they look mainly natural (very snowy winter followed by very wet spring, with deeply frozen soil due to very cold temperatures in the autumn, thus limiting infiltration).

We have revised the manuscript accordingly - please see the response to RC1 #10.

12. Line 259 : " Per assumption c of the index flood method…". More information is needed to understand what assumption c entails. A reference would also be useful.

We have revised the manuscript to clarify assumption c. are copied below:

"With respect to assumption c of the index flood method, which assumes that a regional growth curve can be applied to a homogenous area as outlined above, we attempted to fit a distribution to the ratio of the annual maximum discharge values at each station to the corresponding index flood. Hailegeorgis and Alfredsen (2017) discussed a regionalization procedure which ensures the homogeneity of the station-level data over any region. However, due to the limited availability of the discharge data we avoided such sub-sampling and carried out the index flood method at the entire watershed scale (Faulkner, Warren, & Burn 2016). This, however, has impacted the upper quantiles of the flood estimation when comparing to the station level data."

Please note also that all three assumptions (a,b, and c) were introduced in old manuscript (copied below):

"The index flood approach entails the following assumptions: a) the flood quantiles at any hydrometric site can be segregated into two components – an index flood and regional growth curve (RGC) -; b) the index flood at a given location relates to the (sub)catchment characteristics via a power-scaling equation, either in a simpler case which considers only upstream contributory area or in a more complex case which incorporates land use/ land cover, soil, and climate information; and c) within a homogeneous region the departure/ratio between the index flood and discharge at hydrometric sites yields a single regional growth curve which can relate the discharge and return period (Hailegeorgis & Alfredsen, 2017)"

13. Line 262 : Qi needs to be defined and Qtilde should be more clearly defined as the median annual maximum discharge.

Both Qi and Qtilde were defined within the following excerpt of Section 2.2.2:

" A median annual maximum discharge value was then calculated ($\tilde{Q}$) for each hydrometric station. As discussed in Hailegeorgis & Alfredsen (2017), although the index flood is generally the sample mean of a set of annual maximum discharge values, index floods have also been evaluated based on the sample median (eg. Wilson et al., 2011) at the suggestion of Robson & Reed (1999). Finally, the index flood values ($\tilde{Q}$) were used to normalize the observed annual maximum discharge values (Q) at their respective station ($Q_i = Q/\tilde{Q}$)."

The excerpt has been slightly edited for clarity. The revised text is copied below:

"A median annual maximum discharge value ($\tilde{Q}$, "Qtilde") was then calculated for each hydrometric station. As discussed in Hailegeorgis & Alfredsen (2017), although the index flood is generally the sample mean of a set of annual maximum discharge values, index floods have also

been evaluated based on the sample median (eg. Wilson et al., 2011) at the suggestion of Robson & Reed (1999). Finally, the index flood values ($\tilde{Q}$) were used to normalize the observed annual maximum discharge values (Q) at their respective station, resulting in a set of values designated as Qi, such that Qi = Q/ $\tilde{Q}$."

Additionally, we have modified the following:

"Thus, the regional regression model derived a relationship between index flood ($\tilde{Q}$) and upstream contributory area for each hydrometric station s or sub-catchment outlet. The relationship between index flood at station i or at a subcatchment outlet ($\widetilde{Q^s}$) (median of annual maximum discharge) and upstream contributory area ($A_s$) is given by:

$$\tilde{Q}^s = aA_s^c \quad (1)$$

where $a$ is the index flood discharge response at a unit catchment outlet (or at a hydrometric station) and $c$ is the scaling constant."

14. Lines 298-299: Note that other large-scale flood modelling approaches also don't require channel geometry (e.g. Wing et al. 2017).

2D methods such as LISFLOOD in Wing et. al. 2017 do not require explicit channel geometry specification, rather it is apparent in the input DEM and is resolved by the methods itself, entailing substantial computational power. By "hydraulic methods" we meant typical 1D methods such as the 1-D St. Venant shallow water equation (Brunner, 2016) or the solving of Manning's equation. We have changed the line accordingly. The edited text is copied below:

"Hydraulic methods of discharge calculation typically entail hydraulic parameters derived from the known geometry of a channel. The knowledge of a channel's cross sectional design is a requirement for many one-dimensional flood routing models, for instance the one-dimensional St. Venant equation (Brunner, 2016). The requirement of the cross-section being perpendicular to the flow direction makes it an implicit problem and also dependent on the choice of cross-section position as well as the distance at which the points are taken on the cross-section. In the current practice of hand designing it makes it subjective and draws substantial uncertainty in the inundation simulation. Alternatively, HAND-based models do not explicitly solve the Manning's equation at individual cross-section, but rather solve for a catchment averaged version of it, by considering a river as a summation of infinite cross-sections. As such, the inherent uncertainty is avoided."

15. Lines 311-312: Do you have an uncertainty estimate on slope estimated with the 30m x 30m DEM? As indicated above, having more information on the vertical accuracy of the DEM would be useful.

An uncertainty discussion has been added, in addition to the information on vertical accuracy provided in the response to RC1 #5. The relevant text is copied below:

"The uncertainty in the vertical dimension affects the slopes of individual pixels, the upslope contributing area, and can potentially affect the quality of extracted hydrologic features (Lee et al., 1992, 1996; Liu, 1994; Ehlschlaeger and Shortridge, 1996). Hunter and Goodchild (1997) whilst investigating the effect of simulated changes in elevation at different levels of spatial autocorrelation on slope and aspect calculations, indicated the importance of a stochastic understanding of DEMs. The Monte Carlo method (Fisher, 1991) could potentially shed some light on this kind of uncertainty. However, in our case it was beyond the focus of our study and we assumed the vertical uncertainty is small enough to not affect our large-scale flood modeling simulations."

16. Line 320: Why did you choose the Lotter method? In Tullis (2012), it is stated (p. 72) that "Pillai (1962) concluded that the Horton relationship performed the best and that the Lotter relationship gave inconsistent results." On the same page, Tullis (2012) indicates: "Flintham and Carling (1992) evaluated the Horton, Colebatch, Pavlovskii, and Lotter methods. They concluded that the Pavlovskii relationship was the most accurate, the Horton and Colebatch relationships were satisfactory, and the Lotter relationship performed poorly. Four of the five relationships evaluated in the three different studies were identified at least once as a "best performer," but consensus was not achieved regarding an overall best method. The Lotter relationship, on the other hand, was singled out in each study as "not recommended for use."

It is important to note that, while the Lotter Method was selected for use in the initial manuscript because of its simple but powerful assumption that the total discharge is the sum of all the sub-area discharge (Chow, 1959), the composite Manning's n method can be easily interchanged within the InundatEd system. We agree that the Lotter method was not the best initial selection, but have also established that the selection of the Manning's n method had a very small impact on the final outcome in terms of the median CSIs for our five comparison cases (Grand River watershed RP 100; Ottawa River watershed RP 16.52; Ottawa River watershed RP 25.96; Ottawa River watershed RP 26.5; and Ottawa River watershed RP 42.69).

To demonstrate this, the suggested methods from this comment, and a number of others, were used instead of the Lotter method to generate simulated floods for each of the five comparison cases. These simulated floods were compared to the observed floods as before, and the same metrics were recalculated (including the CSI metric requested in previous comments). Specifically, the following methods were tested: the Colebatch method, the Cox method, the Horton Method, the Krishnamurthy Method, the Pavlovskii Method, and the Yen Method.

Tables are now provided in the Supplementary Materials (Tables S3 and S4, the highest median CSI for each case is highlighted in bold) which give the 25th percentile, median, and 75th percentile CSI values which resulted from each Manning's n method, for each comparison case. The range of CSI values for each metric (25th percentile, median, and 75th percentile) is also included for each comparison case, as evidence of the small impact changing Manning's n method had on the final results. For instance, for the Grand River watershed comparison case, the median CSI value had a range of only 0.016. The ranges of the median CSI values for the four Ottawa River watershed cases are: 0.079, 0.064, 0.022, and 0, for return periods 16.52, 25.96, 26.5, and 42.69 respectively.

In light of these results and the feedback given regarding the Lotter method, we have switched our main method, for the purposes of other reporting and visualizations, to the Krishnamurthy method (Table 4, Figures 6-9). This is now indicated in the manuscript and copied below:

"Of the binary comparison results for the 7 composite Manning's n methods listed in Section 2.2.3, the Krishnamurthy method yielded the highest median CSI values (Table S3 for the Grand River watershed and Table S4 for the Ottawa River watershed). As such, it was selected for further visualization and discussion."

17. Line 325: (Figure 4) This figure has a lot of details which may not be needed, particularly since the HAND method is widely known.

We have substituted a more concise version of Figure 4 (now Figure 2).

18. Line 397: These 2 stations should be identified on Figure 1
The two stations are now identified on Figure 1.

19. Line 403: divided (instead of divded)
This typo has been corrected.
20. Line 447: Are there references related to flood modelling validation for the use of Matthews Correlation Coefficient? The reference cited here (Chicco & Jurman, 2020) is in the Genomics field. The Critical Success Index (CSI) seems more commonly used for flood modelling validation (e.g. the two listed references), so why not use it here to facilitate comparisons with other studies, for example Wing et al. (2017) who obtained a score of 55.2% for their flood maps of the United States.

Although no references for the use of the MCC for flood model validation were included in the manuscript, the MCC has been used in the context of flood model validation. See, for instance, Rahmati et al. (2020) (full citation below), wherein the MCC was used to evaluate the agreement between observed flood data and simulated floods, with respect to flood extent. Another recent example of MCC's use in a flood model validation context can be found in Esfandiari et al. (2020) (full citation below). Additionally, it is important to note that Chicco & Jurman (2020) encourages the use of the MCC, rather than the F1 and Accuracy, "in evaluating binary classification tasks by

all scientific communities", not just by the Genomics field in which the Chicco & Jurman paper was situated (i.e., we see no reason, other than tradition, why the MCC shouldn't be used in this context). However, we agree that the CSI is widely used and thus facilitates result comparisons. As such, we have modified the manuscript to include the CSI in all figures, tables, and results discussions, replacing the metric Accuracy (a known unreliable metric). Additionally, Esfandiari et al. (2020) and Rahmati et al. (2020) have been added to the manuscript.

Esfandiari, M., Abdi, G., Jabari, S., McGrath, H., & Coleman, D. (2020). Flood hazard risk mapping using a pseudo supervised random forest. *Remote sensing, 12 (19), 1-23*. DOI: 10.3390/rs12193206

Rahmati, O., Darabi, H., Panahi, M., Kalantari, Z., Naghibi, S. A., Ferreira, C. S. S., et al. (2020). Development of novel hybridized models for urban flood susceptibility mapping. *Scientific Reports, 10(1), 1-19*. DOI: 10.1038/s41598-020-69703-7.

21. Line 456 (Figure 6): I don't think this figure is needed. Providing HAND results (what are the units on Figure 6a,b?) at the scale of such watersheds is not really useful, and there is also little value to showing the drainage network or the Manning's n values, already presented in a table.

Figure 6 has been removed from the modified manuscript and shifted to supplementary as Figure S3. The unit of HAND is meters.

22. Lines 464-466: "The difference in correlation quality can be accounted for in part by the difference in the relative complexities of the delineated networks of the Grand River and Ottawa River watersheds." This is not a sufficient explanation for such a marked difference in correlation values between the two watersheds. What are the "relative complexities" of the network of the Ottawa River watershed that would explain that several points are not at all following the 1:1 slope?

Please refer to the reply of RC1 # 2.

23. Line 482: The information on dams on the Ottawa and Grand River watersheds should be provided here. To the best of my knowledge, about 40% of the flow is controlled by dams. What is the situation on the Grand River? As indicated below, dam information presented in supplementary material could easily be integrated into Figure 1.

Please refer to the reply of RC1 #10.

24. Lines 490-491: "As expected, for the stations with high observation counts (n = 101 and n = 84 for the Grand River watershed (Figure 8a) and Ottawa River watershed (Figure 8b), respectively) the theoretical and estimated return periods are closer, at least for lower return

periods." I find this sentence confusing. First, it seems to imply that there are stations with lower observation counts, but these are the only two stations presented. Then, the theoretical and estimated return periods are indeed close for low return period, but not at all for longer return period (even when less than the 5T threshold), which seems problematic.

This section has been edited for clarity and discussion regarding the pre-5T differences between the theoretical and estimated quantiles. The two stations (02KF001 for the Ottawa River watershed and 02GA034 for the Grand River watershed) were selected based on their relatively high observation counts. 02GA034 has 46 years of record and 02KF001 has 34. An error in these figures (the count label) has been corrected. The relevant text is copied below:

"The dotted lines on Figures 5a and 5b represent the 5T threshold - the return period limit beyond which flood simulations can not be reasonably estimated. The 5T threshold requires that, for the reasonable estimation of a quantile for a desired return period T, there be at least 5T years of data (Hailegeorgis & Alfredsen, 2017; Jacob et al., 1999). As expected,  the theoretical and estimated return periods are comparable for low return periods. However, and as shown in Figure 5,  the theoretical and estimated quantiles deviate at lower RP values than the 5T threshold for both stations. This disagreement between the theoretical and estimated quantiles recalls the assumption of homogeneity for each watershed (Burn, 1997) - estimations of higher return periods, considering the 5T rule, would require more observations. However, further sub-sampling the stations into regional homogeneous groups would have reduced the data quantity substantially for each group."

25. Lines 497-505: The link between this paragraph and the previous ones is not clear and, overall, this paragraph seems out of place. It is the first time that the hexagonal gridding system is mentioned, and it is difficult to understand why it is problematic. I suggest removing this paragraph.

This paragraph has been removed from the revised manuscript.

26. Lines 518: Why did you not include the CSI, since you are testing 4 metrics?

The previously used indices Accuracy has been replaced in the manuscript by the CSI. CSI was calculated but not included in the visualizations.We have modified the manuscript and results to incorporate CSI, per previous responses.

27. Line 520: (Figure 10): Why is the scale not the same in each of the figures? And why is the area covered for RP 42.69 different from the other maps? Table 4 should be mentioned here, as it is easier to compare with other large-scale flood modelling approaches (e.g. Wing et al. 2017) with actual values than with maps. It would also be easier to use the CSI for this comparison.

The scale differs between the subfigures of Figure 10 (now Figure 7) due to differences in the number and location of the evaluated subcatchments (recorded in Table 4 as "Number of evaluated subcatchments"). The reason for these variations is discussed in Section 2.5.3. In short, only

subcatchments with contributory upstream areas similar to the hydrometric station of interest were included. Additionally, it was necessary that historical data be available for any evaluated subcatchment, in order to evaluate the simulated flood results. As such, certain subcatchments, although they met the upstream area criterion, were not included due to a lack of historical data. Thus, the discrepancy between subfigures a), c), and e) is a result of differences in the availability/ extents of the observed (historical) flood polygons (listed in Table 3). With respect to subfigures g) and h), these are in a different location than subfigures a) - f) since they are centred on a different hydrometric station, necessitating a specific scale bar and north arrow. A reference to Table 4 has been included.

28. Line 527-528 : Wing et al. (2017) also used a 30-m DEM, and Sampson et al. (2015) a 90-m DEM, and they obtained a fit index of 55.2% and 75%, respectively. Perhaps the problem is more related to the HAND approach compared to the hydraulic modelling approach?

We agree a 2D hydraulic modeling approach would yield much more physically justifiable results, especially in the cases of braided rivers with very slight channel slopes. However, a 2D approach requires substantial increases in computational and processing power. We believe that a higher resolution DEM for the areas in the downstream portion of the Ottawa river basin - where the river network connectivity was extremely complex and hard to follow based on single downstream direction assumption-, would have provided a more realistic representation of the system, in turn, a better result. The modified text reads;

"Although the results for both the Grand River watershed and the Ottawa River watershed suggest substantial agreement between the respective observed and simulated flood extents, a number of considerations, including input data characteristics and metric bias, require that the presented results be taken with caution and, in some cases, offer clear paths for improvement. With respect to input data, the simulated floods presented within this case study are limited by the initial use of a 30m x 30 DEM raster. As concluded by Papaioannou et al. (2016), floodplain modelling is sensitive to both the resolution of the input DEM and to the choice of modelling approach. Additionally, and as discussed in Section 2.2.3, there are some inherent limitations of the HAND-based modeling approach."

29. Line 531: Considering that you are not providing a lot of information on the uncertainty in, for example, the slope estimates (see above comment), it remains difficult to be convinced that the problem is with the flood extent polygons. It is also interesting to note that Lim and Brandt (2019), cited here, list CSI in their approaches, but not MCC. Further justification is needed for not using CSI in this study.

Please see the RC1#28. The CSI has been added to all discussions, tables, and results visualizations.

30. Line 545: Again, it is difficult to understand why an index that would allow for comparison with previous studies (such as the CSI) was not used. Providing references where MCC was used to assess the success of simulated floods would be useful.

Please see the above replies, particularly RC1 #20. The CSI index has been added to all discussions, tables, and results visualizations.

31. Line 550: The depth value should still be indicated in the green-red colour scheme on Figure 11. Otherwise, you should use a single fill colour (the same comment applies to Figure 12).

Figures 11 and 12 (now Figures 8 and 9) have been edited to use only a single fill colour as opposed to a gradient, as the focus of these figures is flood extent, not flood depth.

32. Line 554: It seems obvious that a MCC of 0.95 is associated with a strong model success, so I am not sure to understand what is meant here.

The referenced inset figures were included to help the reader to understand and interpret the values of the MCC metric, by visualizing specific simulated vs observed floods. Although we agree that a higher value should be intuitively associated with better model success, we wished to provide evidence beyond a simple accounting of the final MCC values. In the updated Figures 8 and 9, the intention is the same - to aid in understanding and interpretation of our binary classification results and to provide further evidence beyond a simple accounting of the final CSI values.

33. Line 563: The position of dams should be indicated in Figure 1, not in supplementary material.

Figure 1 has been revised to include the location of the dams and other features requested in previous comments..

34. Line 568: It is possible (instead of it's possible)

This typo has been corrected.

35. Line 576: So a single fill colour should be used instead of the green-red legend in Figure 12.

Figures 11 and 12 (now 8 and 9) have been updated to use a single fill colour. Please see the response to RC1 #31 above.

36. Line 593: This should have been mentioned earlier, when presenting Figures 9 and 10.

A reference to Table 4 can now be found at the mentioned position:

"Binary classification results for the Grand River watershed are shown in Figure 6 for four comparison metrics: Critical Success Index, Matthews Correlation Coefficient, True Positive Rate, and True Negative Rate. Figure 7 presents Critical Success Index and Matthews Correlation Coefficient results for the four Ottawa River watershed cases, with True Positive and True Negative results presented in Supplementary Figure S5. Table 4 lists the number of subcatchments evaluated, the median CSI, and the median MCC for each of the 5 test return periods. The median values of additional metrics are provided in Table S5."

37. Lines 605-607: "The moderately high FDR value of 0.44 for the 42.69-year return period and the observed overestimation of flood extent (Figure 12B) may be a result of high local Manning's n values." It is not clear why high Manning's n values would only play a role in this case.

The discussion has been revised in the current manuscript,

"The moderately high FDR value of 0.44 for the 42.69-year return period and the observed overestimation of flood extent (discussed below) may be a result of high local Manning's n values. In addition, the influences of flat terrain (Lim & Brandt, 2019) and anabranch must be considered as it can disrupt the assumption of a single drainage direction for each pixel during sub-catchment delineation. Additional factors potentially influencing the overestimation are the problems inherent to HAND-based modeling, as discussed in section 2.2.3. The topography of the area of the Ottawa River watershed wherein the extent comparisons were made is relatively flat with multiple anabranches and thus can lead to chaotic network delineation."

38. Line 611: relatively (instead of realtively)

This typo has been corrected.

39. Line 613: It is the first time the burning of the polygon network is mentioned.

This information was presented in the GIS pre-processing section (copied below):

"ArcGIS Desktop's Raster Calculator tool was used to burn the river network vector into the DEM to ensure the consistency of the river network between the dem delineated and observed."

40. Supplementary material: Table S1: The number of digits after the decimal point should be consistent and reasonable (e.g. 5 digits for discharge values in m3/s is too many). The same comment applies to Table S2.

The number of decimal points in all tables (main and supplementary) have been reduced and checked for consistency.

41. Line 724: Dalrymple (instead of Darlymple)

This error has been corrected.

RC2:

1. The paper contains much extraneous detail and a number of unnecessary figures, creating a long paper that is difficult to follow in places. Consider which information the reader requires to understand your model, how it works, and how it performs. For instance, equations 2 and 8, figure 6, parts of section 2.2.2 and 3.1.

We removed the figures and equations from the revised manuscript. Also, we have combined other figures together to reduce the figure number.

2. The novel aspects of this framework either do not exist or are inadequately emphasised. The presented RFFA does not appear to be much different from Hailegeorgis & Alfredsen (2017). Much of the Canadian RFFA literature by the likes of Taha Ouarda and Donald Burn is omitted. Advances in large-scale RFFA have been presented in, for instance,

Faulkner et al. (2016, doi:10.1080/07011784.2016.1141665) or Smith et al. (2015, doi:10.1002/2014WR015814) and so the authors should be clear about what is novel about their approach. Similarly, the use of HAND in flood inundation prediction is well documented and so the authors must make clearer what is novel about their approach. Again, key literature on this front such as Afshari et al. (2018, doi:10.1016/j.jhydrol.2017.11.036), Liu et al. (2018, doi:10.1111/1752-1688.12660), and Zheng et al. (2018, doi:10.1111/1752-1688.12661) is missing.

We have expanded our Introduction/ literature review to include the papers mentioned in this comment, and to emphasize the novel aspects of our paper. Overall, the novelty of this research is the application of a big-data, discrete-global grid systems architecture and data model to the context of flood modelling, which expands on the existing methods in the literature by addressing a number of shortcomings and challenges. To that end, this paper introduces a novel, open-source web application (InundatEd) which allows the results of the discrete global grid system-based flood model to be presented in a publicly accessible and open-source format. For further details, please see the response to RC1 #1.

3. The limitations on the functionality of the presented model are inadequately discussed. How does the requirement for quality river gauge data with long records impair the ability to deploy this model at large scales elsewhere? The limitations and inaccuracies of 'planar' models such as HAND are well known, but this is not discussed to any meaningful degree. The suggested literature above, amongst others, shows how physics-lite modelling approaches often correspond poorly with observations of flood inundation.

The long-term flow record from homogeneous stations are essential for design of regional regression models. The unavailability of which affects the flood magnitude computations specifically for the upper quantiles (5T rule). We have included this in our discussion in the revised manuscript (copied below):

"Some of the limitations of this framework include the long-term flow records and homogenous stations required for the creation of regional regression models. A dearth of long-term data affects flood magnitude computations specifically for the upper quantiles (5T rule, Section 3.1)."

With respect to the limitations inherent to the HAND method, copied below:

"However, the simplistic HAND-based model struggles to simulate proper inundation extent in case of complex conditions such as meandering main channels and confluences (Afshari et. al. 2017). This model doesn't capture the dynamic flow characteristics such as backwater effects created by flood mitigation structures. Therefore, users have to be cautious in such cases."

4. The inferences made arising from model validation results are often unsupported. For instance, the wild overprediction in Figure 11b is not unusual for models not grounded in

a derivation of the shallow water equations. Where the benchmark flood is not valley filling and takes place in a wide, flat floodplains – as seems the case in this panel – the failure to simulate the flow of water can often lead to overprediction. Instead, the authors suggest grid resolution may be the issue. I suggest a more in-depth analysis in this section with evidence for the conclusions drawn.

The simplistic HAND-based model, which is not based on derivation from the shallow water equations, struggles to simulate proper inundation extent in case of complex conditions such as meandering main channels and confluences (Afshari et. al. 2017) and leads to overestimation. This model doesn't capture the dynamic flow characteristics such as backwater effects created by flood mitigation structures. In case of 11B (now 8B) the location is both affected by multiple upstream and downstream dams as well as the river appears to be meandering. These are potential causes of the failure of the HAND model and of the unrealistic simulations.

The modified lines are copied below:

"The Grand River watershed yielded a median False Discovery Rate (FDR) of 0.117, and the four Ottawa River watershed cases yielded respective median FDRs of 0.019, 0.01, 0.006, and 0.44 for the evaluated subcatchments. The moderately high FDR value of 0.44 for the 42.69-year return period and the observed overestimation of flood extent (discussed below) may be a result of high local Manning's n values. In addition, the influences of flat terrain (Lim & Brandt, 2019)  and anabranch must be considered as it can disrupt the assumption of a single drainage direction for each pixel during sub-catchment delineation. Additional factors potentially influencing the overestimation are the problems inherent to HAND-based modeling, as discussed in Section 2.2.3."

The relevant excerpt from 2.2.3 is copied below:

"However, the simplistic HAND-based model struggles to simulate proper inundation extent in case of complex conditions such as meandering main channels and confluences (Afshari et. al. 2017). This model doesn't capture the dynamic flow characteristics such as backwater effects created by flood mitigation structures. Therefore, users have to be cautious in such cases."

5.  The lack of a requirement for channel geometry is not clear to me. An understanding of how much flow remains in-channel, which would have no meaningful representation in the DEM, would surely create a much more accurate model. Indeed, I do not know how one can hope to simulate floods such as 1.25, 1.5, 2.0, 2.33 year recurrence (most of which would presumably remain in-bank) without understanding channel conveyance. I think this needs to be further unpacked.

We agree with the reviewer that the in-channel geometry (bathymetry) is not meaningfully represented by a DEM. However, when we talked about channel geometry requirements, we were referring to knowledge of a channel's cross sectional design - a requirement of many one-dimensional flood routing models (eg. one dimension St. Venanet eq.). The requirement of the cross-section being perpendicular to the flow direction makes it an implicit problem and also dependent on the choice of cross-section position as well as the distance at which the points are taken on the cross-section. In the current practice of manually designing it makes it subjective and draws substantial uncertainty in the inundation simulation. Alternatively, HAND-based models do not explicitly solve the Manning's equation at individual cross-section, but rather solve for a catchment averaged version of it, by considering a river as a summation of infinite cross-sections. As such, the inherent uncertainty is avoided.

We agree that the inclusion of in-channel geometry would improve the simulation of the flood, particularly for the low-RP floods where flow would presumably remain in-bank for any flood simulation model. We have updated the discussion to reflect this.

"Hydraulic methods of discharge calculation typically entail hydraulic parameters derived from the known geometry of a channel. The knowledge of a channel's cross sectional design is a requirement for many one-dimensional flood routing models, for instance the one-dimensional St. Venant equation (Brunner, 2016). The requirement of the cross-section being perpendicular to the flow direction makes it an implicit problem and also dependent on the choice of cross-section position as well as the distance at which the points are taken on the cross-section. In the current practice of hand designing it makes it subjective and draws substantial uncertainty in the inundation simulation. Alternatively, HAND-based models do not explicitly solve the Manning's equation at individual cross-section, but rather solve for a catchment averaged version of it, by considering a river as a summation of infinite cross-sections. As such, the inherent uncertainty is avoided."

6. Damage computation is mentioned, but not demonstrated or tested. Consider dropping this component or illustrating a use case – as presently there is no scientific contribution on this front.

Section 2.2.4 (Damage Computation) has been moved to the Supplementary document (Section S1). We agree that it has little value as a scientific contribution, but prefer to keep it as Supplementary information due to its relevance to the functionality of the InundatEd web application.

7. Validation results require much further explanation and contextualisation (groundin in literature). For instance, I have no idea what to take from lines 472-477. The Matthews Correlation Coefficient is not widely used and so makes contextualization impossible. The reader needs more help in understanding what the validation results mean. Equally, it is unclear whether the authors have made errors in their computation or whether the MCC is unfit for purpose: any metric which rewards figure 11b higher than 12b is clearly not doing its job. Consider a more widely used metric so the reader can understand, to some extent,

how your model compares to others in this field. Secondly, consider the purpose of the validation in the context of the purpose of the model. What is the point of your model? What is it meant for? How good does it therefore need to be? If you are computing damages, your benchmark may be higher as this requires accurate depths – if so, test how well the model replicates depths.

To facilitate comparisons, the widely used Critical Success Index (CSI) has been included in all results visualizations, tables, and discussions as recommended in this and previous comments. We agree that the MCC is not widely used in flood literature and that the CSI allows for much better comparisons. The MCC was originally selected due to its robustness against imbalanced classes and its advantages over F1 and Accuracy, as described in Chicco & Jurman (2020). Please see the response to RC1 # 20 for further details and references regarding the use of MCC in the context of flood extents.

With respect to the discrepancy between Figure 11b (now Figure 8b) and Figure 12b (now Figure 9b), a calculation error had been made - we thank the reviewer for bringing this clear mistake to our attention, and have rectified and reviewed accordingly. It may now be seen that Figure 8b - visually a much worse comparison scenario than 9b- yielded CSI values of 0.17 and 0.22. In contrast, Figure 8b yielded a CSI value of 0.66.

Additional discussion has been added which compares the CSI results of this study to the CSI results of others in the field, as well as additional comparisons of the $F_1$ index. The relevant text, starting at Lines (), is copied below.

"The results reported herein are comparable to previously published binary classification values. For instance, Wing et al. (2017) achieved CSI values of 0.552 and 0.504 for a 100-year return period flood model of the conterminous United States at a 30m resolution.
Additionally, the median $F_1$ score (Chicco & Jurman, 2020) for the Grand River watershed was 0.85. The median $F_1$ scores for Ottawa River watershed return periods 26.5, 16.52, 25.96, and 42.69 were 0.96, 0.95, 0.95, and 0.94 respectively. Such results are approximately in line with Pinos & Timbe (2019), who achieved $F_1$ values from 0.625 to 0.941 for 50-year RP floods using a variety of 2D dynamic models. Afshari (2017) achieved $F_1$ values from 0.48 - 0.64 for the 10-year, 100-year, and 500-year return periods when comparing a HAND-based simulation against a HEC-RAS 2D control. Lim & Brandt (2019) which determined that low-resolution DEMs are capable of yielding relatively high comparison metrics (eg $F_1$ values approximately >= 0.80) in situations where Manning's n varies widely over space."

Many areas (in Canada) have no or outdated flood maps and the goal of this study is simple widely available flood inundation mapping through a web interface which can be recomputed on the fly. We lack the necessary datum information for river stages for different gauging stations to properly compare the depths in the river. Also, no flood inundation depths available for comparison for different test cases. We removed the damage computation section from the revised manuscript.

Some of the (necessary) figures require improvement. The colour scales on figures 9 and 10 make it difficult to discriminate 'good' from 'bad'. There is no key on the depth grid for figures 11 and 12, but should just be made a single block colour anyway as this is a binary comparison.

In addition to the inclusion of the CSI in all plots, tables, and discussions, the colour scale of Figures 9 and 10 (now 6 and 7) have been modified for easier viewing and interpretation. Figures 11 and 12 (now 8 and 9) have been modified to use a single block colour instead of a colour gradient.

---

## Referee Report (RR1)

Comments on

"InundatEd-v1.0: A Large-scale Flood Risk Modeling System on a Big-data - Discrete Global Grid System Framework"

Chiranjib Chaudhuri, Annie Gray, and Colin Robertson

The authors have provided detailed responses to reviewers' comments and have made substantial revisions to the paper. The main issue (identified both by myself and the other reviewer) was the lack of novelty of the work. They have clarified the novelty of the proposed approach, which has more to do with managing efficiently big data than with improving large-scale flood modelling per se. Perhaps the title of the paper should be changed to reflect this, i.e. "Dealing more efficiently with big data through the use of Discrete Global Grid System Framework: a case study on flood risk modelling".

There are currently several initiatives in Canada that will result in revised flood maps for very large territories (for example Info-Crue in Quebec which started in 2018 and aims to provide flood maps based on hydraulic modelling for over 25,000 km of rivers by 2023), so it is not clear how this DGGS would be used in practice. Could it serve to store the catalogue of flood simulations that would be produced by each province? HAND flood simulations are useful for visualization purposes, but it is not obvious that they can be described as "reliable flood risk maps" (p. 6, line 153), i.e. the type of maps that can be used in legislation for land-use planning. Furthermore, making flood risk information "more accessible" (p. 6, line 159) is also already achieved in many European countries (e.g. https://flood-warning-information.service.gov.uk/long-term-flood-risk/map) so it is not clear why DGGS is needed to convey this information for the general public. I therefore remain not entirely convinced that there is sufficient novelty to justify a publication.

The Introduction could be shortened by removing detailed information on the impacts of floods and general statements on flood modelling (first two paragraphs). Note that the text specified in the answer to my comments is not the same as what appears in the revised manuscript. For example, on p. 5, line 127, it is stated that "the novelty of this study is twofold", whereas in the response to reviewers, it is stated that "the novelty of this study is threefold". This gives the impression that it was not obvious for the authors to determine what were the novelties in this study…

*Detailed comments*

p. 4, line 95: Afshari et al. should be 2018, not 2017. The reference (p. 30, line 945) lists this paper incorrectly in the alphabetical order as it starts with the first name (Shahab) instead of the surname (Afshari).

p. 6, line 133: Define acronyms the first time they are used (here, RFFA). It is not entirely clear what you mean by "without sacrificing the consistency of the framework". Why would other types of large-scale modelling approach become "inconsistent" if they used either RFFA or HAND (or alternative models)?

p. 6, line 139: A reference is needed to support the statement that "the IDEAS framework uses an integer-based addressing system which makes it orders of magnitude more efficient than that of other, more traditional spatial data models."

p. 8, line 194: "The vertical accuracy of the DEM is 0.34 m ± 6.22 m, i.e., 10 m at the 90% confidence level". Where do these values come from? A reference is needed, as the reported value appears underestimated since it is significantly smaller than what is stated in other publications on SRTM DEMs (e.g. RMSE of 17.76 m in Mukherjee et al., 2013; 13.25 m in Yap et al. 2019). This is important as later you indicate that the vertical uncertainty is "small enough to not affect our large-scale flood modelling simulations". Since LiDAR data are available in several parts of the Ottawa watershed, it would be straightforward to run tests on slope estimated from the SRTM in certain reaches to see how they compare with LiDAR estimates.

p. 8, line 204: As indicated above, comparing slopes obtained by LiDAR and SRTM DEMs over a few reaches would have been relatively simple to do. In fact, vertical accuracy could have been significantly improved by working with a 30-m aggregated LiDAR DEM (where available).

p. 8, line 215: The reference cited here (Comber and Wulder, 2019) doesn't mention Manning's n and thus does not seem appropriate to justify the statement that each pixel is attributed a Manning's n value based on land use/land cover attributes. Considering the uncertainty with HAND, it is not obvious that using a spatially varied Manning's n in the floodplain provides a major advantage over approaches using constant values (e.g. n = 0.035 in the channel, n = 0.1 in the floodplain, Fleishmann et al., 2019).

p. 19, line 531: Are there really braided rivers in the Ottawa watershed? Do you mean in cases where there are islands, resulting in anabranch channels?

p. 21, line 612: Afshari et al. (2018) instead of Afshari (2017).

p. 24, line 693 : Figure S6 (instead of S7). I don't think a figure is needed for this – the fact you had to add 4 seconds to the DGGS makes this figure particularly confusing. The main interest of the proposed methodology is clearly in its efficiency in managing big data, rather than in modelling accurately flood zones, as was pointed out on p. 23 ("InundatEd model allows for the

"swapping" of various flood modelling methods, and thus could easily accommodate, for instance, shallow water equations").

**References**

Fleischmann, A., Paiva, R., & Collischonn, W. (2019). Can regional to continental river hydrodynamic models be locally relevant? A cross-scale comparison. Journal of Hydrology X, 3, 100027, 1-19.

Mukherjee, S., Joshi, P.K., Mukherjee, S., Ghosh, A., Garg, R.D., Mukhopadhyay, A. (2013) Evaluation of vertical accuracy of open source Digital Elevation Model (DEM), International Journal of Applied Earth Observation and Geoinformation, 21, 205-217.

Yap, L.; Kandé, L.H.; Nouayou, R.; Kamguia, J.; Ngouh, N.A.; Makuate, M.B. (2019) Vertical accuracy evaluation of freely available latest high-resolution (30 m) global digital elevation models over Cameroon (Central Africa) with GPS/leveling ground control points. Int. J. Digit. Earth, 12, 500–524.

---

## Author Response (AR2)

**Editor Comments:**

The revised submission is much improved and both reviewers recognise the work done in this regard. The reviewers remain concerned about the novelty, especially in relation to the inundation modelling. Recommendations regarding publication are split, so I have additionally reviewed the manuscript myself and include some comments below. I share the reviewers concern around the novelty of the inundation methods and don't believe the modelling method per say is novel. However, three of the four conclusions relate to the big data architecture and implementation of the modelling methods within this. In that regard I think it is possible for a novel contribution when looking at the work as a whole, with some further revisions to the manuscript needed to bring this out. Please respond to all of the comments below and those from both reviews.

**Comments:**
**1. Line 60-67 need references for examples of these approaches.**

The section is revised to include references. The relevant text (Lines 47-55) and references are copied below

"Flood inundation modelling approaches can be broadly divided into three model classes: empirical (Schumann et al., 2009; Smith, 1997); hydrodynamic (Brunner, 2016, DHI, 2012); and simplified/conceptual (L'homme et al., 2008, Néelz & Pender, 2010). Empirical methods entail direct observation through methods such as remote sensing, measurements, and surveying, and have since evolved into statistical methods informed by fitting relationships to empirical data. Hydrodynamic models, incorporating three subclasses, viz; one-dimensional (Brunner, 2016; DHI, 2003), two-dimensional (DHI, 2012; Moulinec et. al., 2011), and three-dimensional (Prakash et. al., 2014; Vacondio et. al., 2011), consider fluid motion in terms of physical laws to derive and solve equations."

Brunner, G. W. (2016). HEC-RAS River Analysis System 2D Modelling User's Manual Version 5.0. (Report Number CPD-68A). US Army Corps of Engineers Hydrologic Engineering Center.

DHI, 2003. MIKE 11-A Modelling System for Rivers and Channels - User Guide. DHI, p. 430.

DHI. (2012). MIKE 21-2D Modelling of Coast and Sea. DHI Water & Environment Pty Ltd.

L'homme J., P. Sayers, B. Gouldby, P. Samuels, M. Wills, J. (2008) Mulet-Marti Recent development and application of a rapid flood spreading method P. Samuels, S. Huntington, W. Allsop, J. Harrop (Eds.), Flood Risk Management: Research and Practice, Taylor & Francis Group, London, UK,

Moulinec, C., Denis, C., Pham, C.T., Rouge, D., Hervouet, J.M., (2011). TELEMAC: an efficient hydrodynamics suite for massively parallel architectures. Comput. Fluids 51 (1), 30e34.

Prakash, M., Rothauge, K., Cleary, P.W. (2014). Modelling the impact of dam failure scenarios on flood inundation using SPH. Appl. Math. Model. 38 (23), 5515e5534.

Schumann, G., Bates, P.D., Horritt, M.S., Matgen, P., Pappenberger, F., 2009. Progress in integration of remote sensing-derived flood extent and stage data and hydraulic models. Rev. Geophys. 47 (4), RG4001.

Smith, L.C., 1997. Satellite remote sensing of river inundation area, stage, and discharge: a review. Hydrol. Process. 11 (10), 1427e1439

Vacondio, R., Rogers, B., Stansby, P., Mignosa, P., (2011). SPH modeling of shallow flow with open boundaries for practical flood simulation. J. Hydraul. Eng. 138 (6), 530e541

**2. Line 67: needs evidence to support the view that the majority of recent developments for large scale studies are of the simple conceptual type models.**

A reference has been included in the sentence to support the claim. The relevant text (Lines 55-57) and reference are copied below:

"The third model class, simple conceptual, has become increasingly well-known in the contexts of large study areas, data scarcity, and/or stochastic modeling and encompasses the majority of recent developments in inundation modelling practices (Teng et. al. 2017)."

Teng, J., Jakeman, A. J., Vaze, J., Croke, B. F. W., Dutta, D., & Kim, S. (2017). Flood inundation modelling: A review of methods, recent advances and uncertainty analysis. Environmental Modelling and Software, 90, 201–216. https://doi.org/10.1016/j.envsoft.2017.01.006

**3. Line 70: What do you mean by a class of mode in this sentence, please be specific. "A class of model which uses the output of a more complex model as a means of calibrating a relatively simpler model is also gaining popularity (Oubennaceur et al., 2019)."**

By a class of model, we mean the type of models which by construct are simplistic in nature but the data used for calibration of these models comes from more complex modeling studies. For example, Oubennaceur et. al. 2019 used simple power law relationship between discharge and

flood depth in any cell. However, the inundation depth is calibrated based on complex H2D2 model.

**4. Line 81: You are missing a class of parallelisation that uses shared memory threading on CPU's.**

The line has been modified to include the shared memory threading class. The relevant text (Lines 69-72) is copied below:

"With respect to 2D/3D hydrodynamic model code parallelization, Vacondio et al. (2017) listed two approaches: classical (multi-treading or Open Multi-Processing and Message Passing Interface) and Graphics Processing Units (GPUs)."

**5. Line 88-92: very long sentence, consider splitting**

The sentence has been reworked as follows (Lines 91-95):

"Such simple conceptual inundation models offer another potential avenue to handle limitations such as computation requirements and data scarcity. In turn, areas and scales poorly served by standard hydrodynamic modelling may be provided with up-to date flood extent maps. Platforms through which the public can view and interact with the flood extent maps may also be developed (Tavares da Costa, 2019). "

**6. Line 92-101: "One such simple conceptual inundation model…" Given that you are implementing a simple conceptual model this section is very brief. The logic in the introduction breaks down here because you initially state that the simple conceptual models have seen the most progress in recent years and then suggest that conceptual flood models have remained largely uninvestigated. For example why not mention GLOFRIS Assessing flood risk at the global scale: model setup, results, and sensitivity - IOPscience HESS - A framework for global river flood risk assessments (copernicus.org)**
**Or**
**A rapid urban flood inundation and damage assessment model - ScienceDirect**
**I think this is why the reviewers are disputing the novelty claimed here. In my opinion there is a lot of work on simple inundation models that can inform the approach taken here.**

Thanks for the suggestion, we have referenced both of these models in this section as follows in order highlight how our approach is distinct from these. The relevant text (Lines 80-91) is copied below:

"Several studies have introduced generic modelling frameworks that aim to provide robust flood risk estimates with relatively little configuration. Winsemius et al. (2013) for example developed GLOFRIS, a global-scale flood risk modelling framework comprised of global forcing data, a global hydrological model, a flood routing model, and an inundation downscaling model. While capable of providing flood risk at virtually any location on earth, the modelling framework is

fixed to the existing datasets and models used, which have significant uncertainty at the scales considered. At a more local scale, Jamali et al. (2018) introduces a flexible flood inundation model that integrates a 1D hydraulic model with a simple GIS-based flood inundation approach. However, this loosely coupled approach still requires specification of a standalone hydraulic model for each location at which it is implemented. There has been a recent stream of research aiming to develop simple conceptual inundation models that preserve both the generality of GLOFIS and the specificity of more local-scale models."

**7. Line 140: As a non-expert in alternative spatial data models could you state what the traditional models would be in this context. The novelty here was lost on the reviewers and I'm still unsure how significant this is.**

The following text (Lines 143-146) has been added to clarify this and is copied below:

"uses an integer-based addressing system which makes it orders of magnitude more efficient than that of other, more traditional spatial data models (i.e, raster, vector) (Mahdavi-Amiri et. al. 2015; Li & Stefanakis, 2020; Robertson et. al. 2020)."

We have also added the following line to the paragraph where we introduce DGGS (Lines 118-121)

"The Open Geospatial Consortium adopted a DGGS Abstract Specification in late 2017 and work is currently underway to develop standards for DGGS specification as a core geospatial data model (OGC 2017). This is the first use of a DGGS for flood modelling we are aware of."

**8. Line 241: Is it channel or floodplain manning's n being determined? Are both the same? Obviously the link to n is more direct for the floodplain. Channel n is often lower than the floodplain and more associated with channel morphology than adjacent landcover.**

We agree the channel Manning's n can be much lower than the floodplain Manning's n (0.02-0.03). However, the majority of the design flood will inundate the floodplain so we used the floodplain specific LULC dependant Manning's n. Furthermore, for the water pixels on the channel a nominal value of 0.04 has been used.

**9. Line 232: "Regional hydrological frequency analysis at ungauged sites is also studied by few researchers" Is this true? Prediction in ungauged basins is one of the more fundamental and widely studies areas of hydrology. Perhaps this is the case in Canada or there is some more nuance to this point?**

Several studies of the regional hydrological frequency analysis in Canadian context have been done by Prof. D. H. Burn and his group. However, there is in general a scarcity of recent studies on this topic over Canada.

**10. Line 283: check wording of sentence.**

The extraneous "used in" has been removed. The remaining sentence (Lines 290-293) is copied below:

"We took the logarithm of Equation (1) on both sides - a procedure noted in Hailegeorgis & Alfredsen (2017) as used in Eaton, Church, & Ham (2002) - yielding a linear relationship which was solved using the Ordinary Least Squares approach (Haddad et al. (2011))."

**11. Line 294: This is a very long sentence, consider if this can be split and reordered for readability.**

The text has been edited for clarity and is copied below (Lines 302-305):

"A fundamental step of the analysis process is the selection of a suitable probability distribution model, a common tool in hydrologic modelling studies. The model should account for changes to the flow's extreme value characteristics in response to such factors as urbanization, agriculture, resource extraction, or the operation of dams and weirs."

**12. Around line 359: OK 1D hydraulic models require cross-section to be defined. But HAND would also need channel bathymetry to capture the channel flow components of the discharge I assume. How is channel conveyance represented in your HAND method? Is the bathymetry in the DEM and if it was I assume that's been interpolated from cross section data or estimated from a design discharge? So, I agree that on the floodplain HAND avoids the problem of defining cross sections (as a 2D hydrodynamic model also would), but for the channel it will depend on what has been done here and this is not clear to me. I think you oversell the benefits of HAND here.**

We agree the use of bathymetry would improve the reliability of the flood simulation significantly. However, even if bathymetry is presented the 1D model would still need to decide the specific orientation of the cross-section which would have an effect on the simulation results.

We used the interpolated DEM values as the bathymetry of the channel.

We added the following sentence for better clarity in the specific section (Lines 364-370):

"Even though the use of DEM-interpolated bathymetry, as used by our method, induces error in the modelling of flood inundation, it is a necessity in the absence of bathymetry data. There are several instances in literature (Sanders, 2007) where the DEM-interpolated bathymetry has been tested in place of actual bathymetry for hydrodynamic flood modelling. Furthermore, the requirement of the cross-section being perpendicular to the flow direction makes it an implicit problem and also dependent on the choice of cross-section position as well as the distance at which the points are taken on the cross-section."

Sanders, B. F. (2007), Evaluation of on-line DEMs for flood inundation modeling, Advances in Water Resources, Volume 30, Issue 8, 2007, Pages 1831-1843, ISSN 0309-1708, https://doi.org/10.1016/j.advwatres.2007.02.005.

**13. Line 368: Backwater effects are not only due to flood mitigation structures but will occur on all rivers with subcritical flows conditions (which will probably include the vast majority of those with significant floodplains). The issue is more ubiquitous than suggested here.**

We agree the high flow depth and small flow velocities in the natural rivers can cause backwater effect in very far upstream. We revised (Lines 378-382) to include this perspective as follows:

"Furthermore, the large flood depth and low flow velocity in the natural rivers makes the river subcritical on many occasions, specifically for large floodplains where the water slows down significantly. This causes the backwater effect very far upstream of the flooding locations which is not simulated in HAND based methods."

**14. Line 685, sentence needs to be specific regarding what the "direct comparisons" are. I know its obvious from the earlier text but the sentence on its own is truncated in my view.**

The modified sentence reads as follows (Lines 700-702):

"Overall, the results indicated that the current iteration of the InundatEd flood model was reasonably successful on the basis of moderate-high MCC values and direct comparisons against the observed flooding extents."

**15. Line 694: Perhaps I missed it, but I didn't get a good understanding of what a traditional raster based methods are in this context. This links back to earlier comments around line 140 where I didn't really appreciate what the innovation being brought by DGGS was reference to.**

The paragraph is reworded as follows (Lines 710-714):

"There is a distinct contrast of runtimes between the DGGS method and those using a traditional, raster-based method for sub-catchments within the Grand River Watershed (n= 306 for each method) during the generation of respective RP 100 flood maps. The DGGS based storing and processing method is an order of magnitude faster than processing the HAND and catchment boundaries using raster and vector format."

**16. Line 717: I'm happy with principal behind the approach taken to the inundation modelling, but I'm not convinced the use of catchment integrated Manning's is really a novelty of this work. The other novelties listed are stronger in my opinion. Rather than reinvent HAND as novel, would it be more accurate to say an existing method has been**

implemented with a new data set and/or location and computation framework. I don't know what has been done previously in the region, but in my opinion it is the application rather than the inundation modelling that is more likely to be novel. Furthermore, the results show how well the inundation model reproduces historical events and design extents, at no point do you analysis the physical process directly, so I would not claim that the inundation extents are physically justified but instead point to the accuracy of the extents.

The second novelty point is reworked as follows (Lines 735-738);

"Second, the computational framework has been implemented using a regional dataset over locations and at scales which have not been studied before. We successfully demonstrated the merit of the HAND-based inundation modelling to emulate the observed flooding extent for several historical and design floods."

**Referee 1 Comments:**

**The authors are to be commended for their responses to the original reviews and have provided an analysis of model behaviour and set-up that is more robust than the first iteration. I'm sorry to say, however, that the novelty of their contribution is still extremely limited scientifically, and does not significantly further understanding in our field. RFFA+HAND is a model of boundary condition generation and inundation prediction that has been extensively published elsewhere, and this paper does not provide further insight into it. Stated novelties are confined to the model deployment and its architecture, rather than its scientific contribution. The use of LULC data to define Manning's n is not novel. Novelty aside, the modelling process and validation is largely scientifically sound. My only comment is to be careful when comparing skill scores (e.g. CSI) across different models, time, space, and contexts. These are not objective measures of performance, and vary depending on the size of the flood, catchment properties, and quality of the benchmark data. See Mason et al., 2009 (https://doi.org/10.1016/j.jhydrol.2009.02.034); Stephens et al., 2014 (https://doi.org/10.1002/hyp.9979); Wing et al., 2021 (https://doi.org/10.5194/nhess-21-559-2021) for further elaboration on this. The comparison to Wing et al. (2017), for instance, states CSIs of ~0.5 – which isn't strictly true as the coverage of the benchmark data was poor, resulting in a lot of 'non-genuine' overprediction, and thus biasing the results. Metrics in Bates et al. (2021; https://doi.org/10.1029/2020WR028673) account for this, and so should be used for a more reliable CSI comparison.**

The paper introduces the first DGGS-based implementation of a HAND flood inundation model. This is implemented completely in-database which has the following benefits:

- Speed improvement compared to raster-based HAND model .
- On-the-fly computation only requiring the following data inputs: DEM, LULC and stage-discharge curve of the input.

- Scalability - computations done in-database meaning transport of data to application software not required, opening up potential for truly global scale high resolution flood inundation modelling
- Multiscale properties – given the nested nature of DGGS cells visualization of model outputs can be immediately rescaled to any resolution dynamically and/or aggregated as inputs

We have edited the manuscript to highlight its novel contributions throughout.

We have revised the comparison as follow (Lines 616-623)

"Bates et al. (2021) achieved CSI values of 0.69 and 0.82 for a 100-year return period flood model of the conterminous United States at a 30m resolution. It must be noted that direct comparisons between the works listed here and this study must be viewed with caution, due to differences in methodologies, assumptions, data sources, data availability, and return periods between the studies. Furthermore, the extent comparison scores are not necessarily objective measures of performance of the simulation model. They can vary depending on the severity of the flood, catchment characteristics, and quality of the benchmark data (Mason et. al. 2009, Stephens et al., 2014, Wing et. al. 2021)."

**Reviewer 2 Comments:**

**1. The authors have provided detailed responses to reviewers' comments and have made substantial revisions to the paper. The main issue (identified both by myself and the other reviewer) was the lack of novelty of the work. They have clarified the novelty of the proposed approach, which has more to do with managing efficiently big data than with improving large-scale flood modelling per se. Perhaps the title of the paper should be changed to reflect this, i.e. "Dealing more efficiently with big data through the use of Discrete Global Grid System Framework: a case study on flood risk modelling".**

We have revised the title of the paper as follows (Lines 1-2):

InundatEd-v1.0: A HAND-based flood risk modeling system using a Discrete Global Grid System

**2. There are currently several initiatives in Canada that will result in revised flood maps for very large territories (for example Info-Crue in Quebec which started in 2018 and aims to provide flood maps based on hydraulic modelling for over 25,000 km of rivers by 2023), so it is not clear how this DGGS would be used in practice. Could it serve to store the catalogue of flood simulations that would be produced by each province? HAND flood simulations are useful for visualization purposes, but it is not obvious that they can be described as "reliable flood risk maps" (p. 6, line 153), i.e. the type of maps that can be used in legislation for land-use planning. Furthermore, making flood risk information "more accessible" (p. 6, line 159) is also already achieved in many European countries (e.g.**

**https://flood-warninginformation.service.gov.uk/long-term-flood-risk/map) so it is not clear why DGGS is needed to convey this information for the general public. I therefore remain not entirely convinced that there is sufficient novelty to justify a publication.**

There are indeed renewed initiatives to address the dearth of up-to-date flood risk maps in Canada currently underway, including a large amount of work stemming from the Global Water Futures project. However, the type of product as was linked to showing long term flood risk in the UK does not exist in Canada nor will it be the outcome of current efforts to improve flood risk mapping. Further, these maps show precomputed static flood properties, rather than dynamically updated / interactive outputs generated by our system. The novel computational framework we describe enables interactive, generic, large scale implementation of flood risk, something we have not seen elsewhere.

**3. The Introduction could be shortened by removing detailed information on the impacts of floods and general statements on flood modelling (first two paragraphs). Note that the text specified in the answer to my comments is not the same as what appears in the revised manuscript. For example, on p. 5, line 127, it is stated that "the novelty of this study is twofold", whereas in the response to reviewers, it is stated that "the novelty of this study is threefold". This gives the impression that it was not obvious for the authors to determine what were the novelties in this study…**

To reduce the length of the Introduction, we have removed the first paragraph and reduced the length of the second paragraph (Lines 39-46, copied below):

"The practice of flood modelling, which aims to understand, quantify, and represent the characteristics and impacts of flood events across a range of spatial and temporal scales, has long informed the sustainable management of watersheds and water resources including flood risk management (Handmer, 1980; Stevens & Hanschka, 2014; Teng et al., 2017, 2019; Towe et al., 2020). Flood modelling research has increased in response to such factors as predicted climate change impacts (Wilby & Keenan, 2012) and advancements in computer, GIS (Geographic Information Systems), and remote sensing technologies, among others (Kalyanapu, Shankar, Pardyjak, Judi, & Burian, 2011; Vojtek & Vojteková, 2016; Wang & Cheng, 2007)."

Regarding the discrepancy between the manuscript and reply, we apologise for the incorrect update of the reply. We combined the first two point which involves both DGGS and big-data architecture into a single point in the final manuscript. However, we made a mistake when we updated the reply document.

**Comments:**

**4. p. 4, line 95: Afshari et al. should be 2018, not 2017. The reference (p. 30, line 945) lists this paper incorrectly in the alphabetical order as it starts with the first name (Shahab) instead of the surname (Afshari).**

Corrected in the revised manuscript.

**5. p. 6, line 133: Define acronyms the first time they are used (here, RFFA). It is not entirely clear what you mean by "without sacrificing the consistency of the framework". Why would other types of large-scale modelling approach become "inconsistent" if they used either RFFA or HAND (or alternative models)?**

The acronym of Regional Flood Frequency Analysis (RFFA) has been updated (Line 138).

By "consistency" we meant that the inundation modeling modules doesn't depend on the earlier discharge estimation modules and vice verse. Its possible to replace with another model without breaking the entire framework. In other methods, such as 1-D shallow water equation, the equation can be solved as a routing method for discharge inside the discharge estimation module. However, that type of approach makes it very hard to separate out one module from another.

**6. p. 6, line 139: A reference is needed to support the statement that "the IDEAS framework uses an integer-based addressing system which makes it orders of magnitude more efficient than that of other, more traditional spatial data models."**

We have added 3 references to support this claim. The relevant text (Lines 143-146) and references are copied below:

"In terms of the tradeoff between model complexity and computation power, the IDEAS framework uses an integer-based addressing system which makes it orders of magnitude more efficient than that of other, more traditional spatial data models (i.e, raster, vector) (Mahdavi-Amiri et. al. 2015; Li & Stefanakis, 2020; Robertson et al., 2020)."

Mahdavi-Amiri, A., Alderson, T., & Samavati, F. (2015) , A Survey of Digital Earth, Computers & Graphics, Volume 53, Part B, Pages 95-117, ISSN 0097-8493, https://doi.org/10.1016/j.cag.2015.08.005.

Li, M., Stefanakis, E. (2020) Geospatial Operations of Discrete Global Grid Systems—a Comparison with Traditional GIS. *J geovis spat anal* **4,** 26. https://doi.org/10.1007/s41651-020-00066-3

Robertson, C., Chaudhuri, C., Hojati, M., & Roberts, S. (2020). An integrated environmental analytics system (IDEAS) based on a DGGS. *ISPRS Journal of Photogrammetry and Remote Sensing, 162,* 214-228.

**7. p. 8, line 194: "The vertical accuracy of the DEM is 0.34 m ± 6.22 m, i.e., 10 m at the 90% confidence level". Where do these values come from? A reference is needed, as the reported value appears underestimated since it is significantly smaller than what is stated in other publications on SRTM DEMs (e.g. RMSE of 17.76 m in Mukherjee et al., 2013; 13.25 m in Yap et al. 2019). This is important as later you indicate that the vertical**

**uncertainty is "small enough to not affect our large-scale flood modelling simulations". Since LiDAR data are available in several parts of the Ottawa watershed, it would be straightforward to run tests on slope estimated from the SRTM in certain reaches to see how they compare with LiDAR estimates.**

The vertical accuracy values are sourced jointly from the DEM metadata, the source webpage at the open.canada.ca website (Canada Centre for Mapping and Earth Observation (2015)) and the publication Beaulieu & Clavet (2007) on behalf of Natural Resources Canada (the data provider). The latter has been added to the manuscript. The revised text (Lines 201-202) and new reference are copied below:

"The vertical accuracy of the DEM is 0.34 m ± 6.22 m, i.e., 10 m at the 90% confidence level (Beaulieu & Clavet, 2007)."

Beaulieu, A., & Clavet, D. (2007). Accuracy Assessment of Canadian Digital Elevation Data using ICESat. *Photogrammetric Engineering & Remote Sensing, 75(1),* 81-86.

We did some comparison of river slope of 44 catchments against the 1m lidar redrived DTM. The comparison is reasonably consistent with a correlation of 0.83 (please see the figures below).

**Figure 1: The location of the tested catchments.**

[Figure]

**Figure 2: Comparison of the river slope from 30m CDEM and derived from 1m LiDAR data.**

[Figure]

**8. p. 8, line 204: As indicated above, comparing slopes obtained by LiDAR and SRTM DEMs over a few reaches would have been relatively simple to do. In fact, vertical accuracy could have been significantly improved by working with a 30-m aggregated LiDAR DEM (where available).**

We understand that use of high-resolution DEM or aggregated high-resolution DEM can potentially improve the quality of the simulations. However, merging two differently processed DEM creates additional problems during delineation. Also, we wanted to be consistent in our entire method with a single dataset. Therefore, we used the CDEM 30m dataset.

**9. p. 8, line 215: The reference cited here (Comber and Wulder, 2019) doesn't mention Manning's n and thus does not seem appropriate to justify the statement that each pixel is attributed a Manning's n value based on land use/land cover attributes. Considering the uncertainty with HAND, it is not obvious that using a spatially varied Manning's n in the floodplain provides a major advantage over approaches using constant values (e.g. n = 0.035 in the channel, n = 0.1 in the floodplain, Fleishmann et al., 2019).**

The reference (Line 223) is replaced with

Brunner, G. W. (2016). *HEC-RAS River Analysis System 2D Modelling User's Manual Version 5.0.* (Report Number CPD-68A). US Army Corps of Engineers Hydrologic Engineering Center.

We agree that the use of spatially varied Manning's n would have marginal effect on the simulation quality in contrast to use static Manning' n values for both channels and flood plain.

However, the forests, which account for the majority of the Ottawa River watershed's land (~73% on the Quebec side) (Table 1, Environment and Climate Change Canada, 2019), may have higher roughness than 0.1 (we have used 0.16). In a large watershed it may be worthwhile to use the spatially varied Manning' n. Fleishmann et al., 2019 have used parameterized rectangular cross-section but in our case we have used natural channel geometry itself. It may not be an exact comparison.

**10. p. 19, line 531: Are there really braided rivers in the Ottawa watershed? Do you mean in cases where there are islands, resulting in anabranch channels?**

Yes, we meant at the downstream of the Ottawa watershed where there are several islands and anabranches in the channel.

**11. p. 21, line 612: Afshari et al. (2018) instead of Afshari (2017).**

The reference is corrected in the revised manuscript.

**12. p. 24, line 693 : Figure S6 (instead of S7). I don't think a figure is needed for this – the fact you had to add 4 seconds to the DGGS makes this figure particularly confusing. The main interest of the proposed methodology is clearly in its efficiency in managing big data, rather than in modelling accurately flood zones, as was pointed out on p. 23 ("InundatEd model allows for the "swapping" of various flood modelling methods, and thus could easily accommodate, for instance, shallow water equations").**

The Figure S6 has been removed from the supporting material. The associated text has been edited to describe the runtime comparison without reference to Figure S6 and is copied below (Lines 710-716):

"There is a distinct contrast of runtimes between the DGGS method and those using a traditional, raster-based method for sub-catchments within the Grand River Watershed (n= 306 for each method) during the generation of respective RP 100 flood maps. The DGGS based storing and processing method is order of magnitude faster than processing the HAND and catchment boundaries using raster and vector format. The mean runtime using the DGGS method (0.23 seconds) was significantly lower than the mean runtime using the raster-based method (3.98 seconds) at both the 99% confidence intervals (p < 2.2e-16)."

---

## Author Response (AR3)

**Editor Comment:**

1. **The text on line 385 (tracked document) is much improved, but the real issue with using interpolated bathymetry as describe here is that the channel bed is not observed and the overall conveyance of the river is biased towards underestimation as a result. You assume this is negligible, which might be the case, but its an assumption that should be clear.**

We have added a sentence to reflect this fact (Line no. 366-367s).

"We assumed the underestimation of channel conveyance due to the use of interpolated DEM bathymetry to be negligible."